# Incorporating Oxygen Isotopes of Oxidized Reactive Nitrogen in the Regional Atmospheric Chemistry Mechanism, Version 2 (ICOIN-RACM2)

Wendell W. Walters[1,2,3], Masayuki Takeuchi[4], Nga L. Ng[4,5,6], and Meredith G. Hastings[2,3]

[1]Department of Chemistry and Biochemistry, University of South Carolina, 631 Sumter St, Columbia, SC 29208, USA
[2]Institute at Brown for Environment and Society, Brown University, 85 Waterman St, Providence, RI 02912, USA
[3]Department of Earth, Environmental, and Planetary Sciences, Brown University, 324 Brook Street, Providence, RI 02912, USA
[4]School of Civil and Environmental Engineering, Georgia Institute of Technology, 311 Ferst Drive NW, Atlanta, GA 30332, USA
[5]School of Chemical and Biomolecular Engineering, Georgia Institute of Technology, 311 Ferst Drive NW, Atlanta, GA 30332, USA
[6]School of Earth and Atmospheric Sciences, Georgia Institute of Technology, 311 Ferst Drive NW, Atlanta, GA 30332, USA

**Correspondence:** Wendell W. Walters (wendellw@mailbox.sc.edu)

**Abstract.** The oxygen isotope anomaly ($\Delta^{17}O = \delta^{17}O - 0.52 \times \delta^{18}O > 0$) has proven to be a robust tool for probing photochemical cycling and atmospheric formation pathways of oxidized reactive nitrogen ($NO_y$). Several studies have developed modeling techniques to implicitly model $\Delta^{17}O$ of $NO_y$ molecules based on numerous assumptions that may not always be valid. Thus, these models may be oversimplified and limit our ability to compare model $\Delta^{17}O$ values of $NO_y$ with observations. In this

work, we introduce a novel method for explicit tracking $\Delta^{17}O$ transfer and propagation into $NO_y$ and odd oxygen ($O_x$), integrated into the Regional Atmospheric Chemistry Mechanism, version 2 (RACM2). Termed ICOIN-RACM2 (InCorporating Oxygen Isotopes of $NO_y$ in RACM2), this new model includes the addition of 55 new species and 729 replicate reactions to represent the propagation of $\Delta^{17}O$ derived from $O_3$ into $NO_y$ and $O_x$. Employing this mechanism within a box model, we simulate $\Delta^{17}O$ for various $NO_y$ and $O_x$ molecules for chamber experiments with varying initial nitrogen oxides ($NO_x = NO + NO_2$)

and $\alpha$-pinene conditions, revealing response shifts in $\Delta^{17}O$ linked to distinct oxidant conditions. Furthermore, diel cycles are simulated under two summertime scenarios, representative of an urban and rural site, revealing pronounced $\Delta^{17}O$ diurnal patterns for several $NO_y$ components and substantial $\Delta^{17}O$ differences associated with pollution levels (urban vs. rural). Overall, the proposed mechanism offers the potential to assess $NO_y$ oxidation chemistry in chamber studies and air quality campaigns through $\Delta^{17}O$ model comparisons against observations. The integration of this mechanism into a 3-D atmospheric chemistry

transport model is expected to notably enhance our capacity to model and anticipate $\Delta^{17}O$ across landscapes, consequently refining model representations of atmospheric chemistry and tropospheric oxidation capacity.

# 1 Introduction

Nitrogen oxides ($NO_x$ = NO + $NO_2$) are essential trace gases primarily released through human activities, carrying significant implications for air quality, nutrient deposition, and the climate system (Galloway et al., 2004; Pinder et al., 2012). $NO_x$ directly modulates atmospheric oxidation processes, consequently impacting the concentrations of various trace gases, including greenhouse gases (Prinn, 2003). Ultimately, $NO_x$ is removed from the atmosphere as atmospheric nitrate. This global process is dominated by the formation of inorganic nitrate, encompassing nitric acid ($HNO_3$) and particulate nitrate ($pNO_3$) (Alexander et al., 2020), although the generation of organic nitrates ($RONO_2$) might be significant in remote and rural areas (Browne and Cohen, 2012). However, both $pNO_3$ and $RONO_2$ may not be a terminal sink for $NO_x$ due to the potential for renoxification from photolysis (Wang et al., 2023; Gen et al., 2022). Uncertainties surrounding the rate of $NO_x$ oxidation to atmospheric nitrate constitute a substantial source of ambiguity in models, influencing ozone ($O_3$) and hydroxyl radical (OH) formation, with important implications for greenhouse gas removal rates (Newsome and Evans, 2017).

The oxygen mass-independent fractionation leading to an oxygen isotope anomaly ($\Delta^{17}O = \delta^{17}O - 0.52 \times \delta^{18}O > 0$) has emerged as a potent tool for evaluating the photochemical cycling and oxidation chemistry of $NO_x$ and its oxidized products ($NO_y$ = $NO_x$ + $HNO_3$ + $RONO_2$ + nitrous acid (HONO) + peroxyacetyl nitrate (PAN) + etc) (Alexander et al., 2020, 2009; Hastings et al., 2003; Michalski et al., 2003; Morin et al., 2011; Walters et al., 2019). While several atmospheric reactions can induce oxygen mass-independent fractionation (Röckmann et al., 1998; Velivetskaya et al., 2016), $O_3$ is the overwhelming source of mass-independent fractionation in the lower atmosphere, which derive from unconventional isotope effects during its formation (Gao and Marcus, 2001). In this work, we focus on the propagation of the oxygen isotope anomaly from $O_3$ mass-independent fractionation into $NO_y$ and $O_x$ molecules for applications to the lower atmosphere. The $\Delta^{17}O(O_3)$ has been measured to be between 20 and 46 ‰ (Krankowsky et al., 2000; Mauersberger et al., 2001). This range of values has been shown to track with the pressure and temperature associated with $O_3$ formation (Thiemens and Jackson, 1990; Morton et al., 1990). For typical tropospheric conditions, $O_3$ exhibits a $\Delta^{17}O$ between 20 and 30 ‰ (Johnston and Thiemens, 1997), with recent near-surface observations suggesting a mean $\Delta^{17}O(O_3)$ near 26 ‰ (Vicars and Savarino, 2014; Vicars et al., 2012; Ishino et al., 2017). $O_3$ is also isotopically asymmetrical such that the $\Delta^{17}O$ of its terminal and central O atoms are different (Janssen, 2005; Marcus, 2008). This intramolecular $\Delta^{17}O$ distribution is significant because the terminal O-atom of $O_3$ (defined as $O_3^{term}$) is preferentially transferred during oxidation reactions involving $O_3$ (Bhattacharya et al., 2008; Liu et al., 2001; Michalski and Bhattacharya, 2009; Walters and Michalski, 2016). The relationship between $\Delta^{17}O(O_3)$ and $\Delta^{17}O(O_3^{term})$ is complex, though experimental data has suggested the following relationship:

$$\Delta^{17}O(O_3^{term}) = 1.5 \times \Delta^{17}O(O_3) \tag{1}$$

Applying this relationship to the assumed tropospheric mean $\Delta^{17}O(O_3)$ of 26 ‰ would imply a $\Delta^{17}O(O_3^{term})$ of 39 ‰, which is near the average of recent near-surface $\Delta^{17}O(O_3^{term})$ observations of 39.3±2 ‰ (Vicars and Savarino, 2014). It is important to note that there could be seasonal differences in $\Delta^{17}O(O_3^{term})$ as inferred from $\Delta^{17}O$ measurements of nitrate at Dome C

(Savarino et al., 2016). On the other hand, direct observations of $\Delta^{17}O(O_3^{\text{term}})$ have reported insignificant seasonal variability

at Dumont d'Urville (Ishino et al., 2017). Stratospheric intrusion events could introduce $O_3$ with an elevated $\Delta^{17}O(O_3^{\text{term}})$ due to higher stratosphere values relative to the troposphere (Krankowsky et al., 2007). Nevertheless, a recent modeling study of $\Delta^{17}O$ of atmospheric nitrate indicated that an assumed $\Delta^{17}O(O_3^{\text{term}})$ value of 39 ‰, reasonably reproduced global tropospheric observations (Alexander et al., 2020). Further, recent chamber simulations have reported a $\Delta^{17}O(NO_2)$ that reached as high as 40.1 ‰ (Blum et al., 2023), which is within the measurement uncertainty of the assumed $\Delta^{17}O(O_3^{\text{term}})$ value of 39.3 $\pm 2$

55   ‰, assuming $NO_2$ formation to be dominated by NO reaction with $O_3$. Thus, while there may be some unresolved uncertainty regarding the $\Delta^{17}O(O_3^{\text{term}})$ value, an assumed tropospheric average of 39.3$\pm 2$ ‰, should reasonably approximate $\Delta^{17}O$ propagation into $NO_y$ molecules in the lower troposphere. In contrast, most other oxygen-bearing atmospheric molecules, such as oxygen ($O_2$), water ($H_2O$), and peroxy radicals ($RO_2$ or $HO_2$), possess (or expected to possess) $\Delta^{17}O$ values near 0 ‰ (Lyons, 2001). These large $\Delta^{17}O$ differences enable the quantitative tracking of the influence of $O_3$ in $NO_x$ oxidation chemistry.

Past observations of $\Delta^{17}O$ in atmospheric nitrate, which includes $HNO_3$, $pNO_3$, and wet-deposited nitrate ($NO_3^-{}_{(aq)}$), have generally shown marked seasonal variations, reflecting shifts between $O_3$ and $HO_x$ chemical regimes influencing $NO_x$ photochemical cycling and atmospheric nitrate production (Kim et al., 2023; Michalski et al., 2012, 2003). However, harnessing the full diagnostic potential of $\Delta^{17}O$ observations necessitates a model framework that can accurately assess and refine the representation of nitrate chemistry while linking it to nitrogen deposition and air quality. Several 0-D box models and a single

3-D global atmospheric chemistry model have been developed to simulate $\Delta^{17}O$ (Michalski et al., 2003; Morin et al., 2011; Alexander et al., 2020, 2009). These models often rely on implicit tagging of $NO_2$ and $HNO_3$ production rates, underpinned by assumptions regarding oxygen-isotope mass-balance calculations, $NO_x$ photochemical cycling dynamics, and $\Delta^{17}O$ values of reactive oxygen species ($O_x$).

While most existing $\Delta^{17}O$ measurements pertain to atmospheric nitrate from deposition and filter samples, our capability to

measure $\Delta^{17}O$ in other $NO_y$ molecules has been rapidly expanding (Albertin et al., 2021; Blum et al., 2023). Based on oxygen isotope mass-balance principles, substantial $\Delta^{17}O$ variations are anticipated among different $NO_y$ including $NO_2$, HONO, peroxy nitrates ($RO_2NO_2$), organic nitrates ($RONO_2$), and $HNO_3$, contingent upon their formation pathways (Table 1). These mass-balance considerations necessitate precise knowledge of the $\Delta^{17}O$ values for several $NO_y$ and $O_x$ molecules. Conventional model approaches have assumed that $\Delta^{17}O$ of OH, $RO_2$, and $HO_2$ are approximately equal to 0 ‰, due to water vapor

isotope exchange or transfer of O-atoms from atmospheric $O_2$(Michalski et al., 2012; Barkan and Luz, 2003). However, some of these assumptions are not valid for all relevant atmospheric conditions, such as under low relative humidity and high $NO_x$ conditions, in which the chemical reactivity of OH could be higher than its chemical lifetime to achieve isotope equilibrium with $H_2O$ (Michalski et al., 2012). Further, $\Delta^{17}O(NO)$ is commonly assumed to be equal to $\Delta^{17}O(NO_2)$ due to their rapid photochemical cycling, such that the $\Delta^{17}O$ values of NO and $NO_2$ reflect the relative contributions of the oxidants involved

in $NO_x$ photochemical cycling (Alexander et al., 2020, 2009; Michalski et al., 2003; Morin et al., 2011). However, recent diel observations of $\delta^{18}O(NO_2)$ (which tracks with $\Delta^{17}O$) and $\Delta^{17}O(NO_2)$ reveal that this assumption is not universally valid due to substantial nocturnal NO emissions close to the surface (Walters et al., 2018; Albertin et al., 2021). The freshly emitted NO,

with a presumed $\Delta^{17}O$ of 0 ‰, would dilute the residual $\Delta^{17}O$ of $NO_x$ from the daytime. The nocturnal primary emissions of $NO_y$ components, including NO, $NO_2$, and HONO significantly impacts our ability to model $\Delta^{17}O$ using implicit methods in polluted regions, employing prior modeling techniques and oxygen isotope mass-balance calculations. This modeling limitation presently impedes our capacity to leverage models for comparison with $\Delta^{17}O$ observational constraints quantitatively to improve understanding of regional and global $NO_x$ oxidation chemistry.

This study is dedicated to addressing uncertainties in modeling $\Delta^{17}O$ for various $NO_y$ molecules. We introduce a novel gas-phase chemical mechanism, designated "InCorporating Oxygen Isotopes of $NO_y$ in RACM2," built upon the foundation of the Regional Atmospheric Chemistry Model, Version 2 (RACM2) (Goliff et al., 2013). This innovative mechanism explicitly traces the transfer and propagation of $\Delta^{17}O$ from $O_3$ into $NO_y$ molecules, with important future implications for chamber experiments and air quality studies.

## 2   Methods

### 2.1   ICOIN-RACM2 Description

The ICOIN-RACM2 mechanism was based on the widely used RACM2 gas-phase chemical mechanism framework (Goliff et al., 2013). The RACM2 mechanism was developed to be able to simulate remote to polluted conditions from the surface to the upper troposphere. The mechanism includes 46 reactions to represent inorganic chemistry. The mechanism aggregates organic reactions based on the magnitude of emission rates, similarities in functional groups, and the compounds' reactivity (Stockwell et al., 1997) and consists of 54 stable organic species, 42 organic intermediates, 317 reactions, including 24 photolysis reactions. Overall, the RACM2 simulated concentrations of gas-phase products compare favorably to environmental chamber data (Goliff et al., 2013).

To simulate $\Delta^{17}O$ in various $NO_y$ and $O_x$ molecules, the transfer and propagation of the oxygen isotope anomaly deriving from $O_3$ were explicitly modeled in the employed chemical mechanism. Previously, a study developed a kinetic model that explicitly tracks the $^{16}O$, $^{17}O$, and $^{18}O$ abundance involving $NO_x/O_3/O_2$ reactions (Michalski et al., 2014). Here, we have adapted and simplified this model framework to explicitly track the transfer and propagation of O atoms derived from the terminal end of $O_3$ without simulating and tracking the absolute $^{16}O$, $^{17}O$, and $^{18}O$ abundances, which can be tedious to employ in a detailed chemical mechanism. Our approach tagged O atoms transferred from $O_3$ as "Q" and tracked the interactions and propagation of "Q" among $NO_y$ and $O_x$ isotopologues using mass-balance and considering isotopologue reaction stoichiometry. The tagging of O isotopologues was not conducted for large O-reservoirs, including $O_2$ and $H_2O$. The reaction mechanism involved reactions with a single tagged O isotopologue, in which one tagged O isotopologue compound was found in the reactant and product, for example, $NO + O_3 \rightarrow NOQ + O_2$ and $NQ + O_3 \rightarrow NQ_2 + O_2$. Additionally, the mechanism involved reactions containing multiple tagged O compounds in the reactants and products, for example, $NQ + NO_3 \rightarrow NOQ + NO_2$. For these multiple O-tagged isotopologue reactions, statistical probabilities, and mass-balance was considered in the product distributions. The

**Table 1.** Summary of the major formation pathways of several $NO_y$ components, reaction types, and their expected $\Delta^{17}O$ values based on oxygen isotope mass-balance. X refers to halogens (Br, Cl, and I) and HC refers to hydrocarbons.

| Formation Pathway | Type | Expected $\Delta^{17}O$ |
|---|---|---|
| | *NO₂* | |
| NO + O₃ | Homogeneous | $\frac{1}{2}(\Delta^{17}O(NO)) + \frac{1}{2}(\Delta^{17}O(O_3{}^{term}))$ |
| NO + RO₂ | Homogeneous | $\frac{1}{2}(\Delta^{17}O(NO)) + \frac{1}{2}(\Delta^{17}O(RO_2))$ |
| NO + HO₂ | Homogeneous | $\frac{1}{2}(\Delta^{17}O(NO) + \frac{1}{2}(\Delta^{17}O(HO_2))$ |
| NO + XO | Homogeneous | $\frac{1}{2}(\Delta^{17}O(NO)) + \frac{1}{2}(\Delta^{17}O(XO))$ |
| | *HONO* | |
| NO + OH | Homogeneous | $\frac{1}{2}(\Delta^{17}O(NO)) + \frac{1}{2}(\Delta^{17}O(OH))$ |
| NO₂ + H₂O(aq) | Heterogeneous | $\Delta^{17}O(NO_2)$ |
| pNO₃ + hν | Heterogeneous | $\Delta^{17}O(pNO_3)$ |
| | *RO₂NO₂*[a] | |
| NO₂ + RO₂ | Homogeneous | $\frac{2}{3}(\Delta^{17}O(NO_2)) + \frac{1}{3}(\Delta^{17}O(RO_2))$ |
| | *RONO₂* | |
| NO + RO₂ | Homogeneous | $\frac{1}{3}(\Delta^{17}O(NO)) + \frac{2}{3}(\Delta^{17}O(RO_2))$ |
| NO₃ + HC | Homogeneous | $\Delta^{17}O(NO_3)$ |
| | *HNO₃* | |
| NO₂ + OH | Homogeneous | $\frac{2}{3}(\Delta^{17}O(NO_2)) + \frac{1}{3}(\Delta^{17}O(OH))$ |
| NO₃ + HC | Homogeneous | $(\Delta^{17}O(NO_3))$ |
| NO + HO₂ | Homogeneous | $\frac{1}{3}(\Delta^{17}O(NO)) + \frac{2}{3}(\Delta^{17}O(HO_2))$ |
| N₂O₅ + H₂O(aq) | Heterogeneous | $\frac{5}{6}(\Delta^{17}O(N_2O_5)) + \frac{1}{6}(\Delta^{17}O(H_2O))$ |
| XNO₃ + H₂O(aq) | Heterogeneous | $(\Delta^{17}O(XNO_3))$ |
| NO₂ + H₂O(aq) | Heterogeneous | $\frac{2}{3}(\Delta^{17}O(NO_2)) + \frac{1}{3}(\Delta^{17}O(H_2O))$ |
| NO₃ + H₂O(aq) | Heterogeneous | $(\Delta^{17}O(NO_3))$ |
| RONO₂ + H₂O(aq) | Heterogeneous | $(\Delta^{17}O(RONO_2))$ |

[a] $\Delta^{17}O$ calculated from the nitrooxy (-NO₃) functional group

explicit tracking and propagation of "Q" in the ICOIN-RACM2 lead to the rename of 19 reactions, the addition of 729 reactions

replicated for the considered O isotopologues, and the addition of 55 oxygen isotopologues of $NO_y$ and $O_x$ relative to RACM2. Additionally, 26 oxygen isotope exchange reactions were added to the ICOIN-RACM2 chemical mechanism (Lyons, 2001) (Table 2).

Based on the model output of the concentrations of the oxygen isotopologues, the $\Delta^{17}O$ of various $NO_y$ and $HO_x$ molecules were calculated, as the following (Eq. 2):

$$\Delta^{17}O(X) = f(Q) \times \Delta^{17}O(O_3{}^{\text{term}}) \tag{2}$$

where $X$ refers to the various $NO_y$ and $O_x$ molecules and $f(Q)$ is the fractional amount of O-atoms deriving from $O_3$ for a particular molecule (i.e., the fractional amount of "Q" atoms). The $\Delta^{17}O(O_3{}^{\text{term}})$ represents the $\Delta^{17}O$ value of the terminal and transferable O atom of $O_3$. For the demonstration of the developed mechanism for applications to chamber simulations and tropospheric chemistry, we have utilized a constant $\Delta^{17}O(O_3{}^{\text{term}})$ value of 39.3±2 ‰, based on near surface-level collections

of $O_3$ on a nitrite coated filter (Vicars and Savarino, 2014; Ishino et al., 2017). This $\Delta^{17}O(O_3{}^{\text{term}})$ value was recently utilized in the global modeling of $\Delta^{17}O$ of atmospheric nitrate, demonstrating reasonable agreement between model simulation and observations of tropospheric nitrate (Alexander et al., 2020). The $\Delta^{17}O(O_3{}^{\text{term}})$ could have temporal variability as well as be influenced by stratospheric intrusion events, which could introduce $O_3$ with a higher $\Delta^{17}O(O_3{}^{\text{term}})$ value. The developed model framework is highly flexible and the user may apply a different $\Delta^{17}O(O_3{}^{\text{term}})$ than chosen for our model simulations,

which will allow users to investigate both the chemical and $\Delta^{17}O(O_3{}^{\text{term}})$ variability on $\Delta^{17}O$ of $NO_y$ and $O_x$ species when interpreting field observations. The $f(Q)$ for the various considered molecules is calculated as followed (Eq. 3):

$$f(Q, X) = \frac{\sum_{i=1}^{j} i \cdot [Z_{\text{with iQ}}(X)]}{\sum_{i=0}^{j} j \cdot [Z(X)]} \tag{3}$$

where $Z$ represents the oxygen isotopologues of molecule $X$, which can contain $Q$ ($Z_{\text{with iQ}}$) or not, $i$ represents the number of $Q$ isotopes present in each $Z$ of $X$, and $j$ represents the maximum number of $Q$ isotopes that can exist in $X$. Overall, this equation

considers the distribution of $Q$ isotopes within different arrangements and calculates the fraction of $Q$ isotopes in the molecule relative to the total number of oxygen atoms. While our mechanism and application is focused on evaluating the propagation of oxygen isotope mass-independent fractionation from $O_3$ into $NO_y$ and $O_x$, the model could be adapted for tracking other potential oxygen mass-independet fractionation, such as $HO_2 + HO_2$ or $CO + OH$ reactions (Röckmann et al., 1998; Velivetskaya et al., 2016), by adjusting the product distribution of "Q" and "O", such that the fraction of "Q" once scaled by the

chosen $\Delta^{17}O(O_3{}^{\text{term}})$ value would match the intended $\Delta^{17}O$ value associated with the oxygen mass-independent fractionation. Previous experiments have reported an increase in $\Delta^{17}O(H_2O_2)$ as the initial $O_2$ concentrations increased (Velivetskaya et al., 2016). This result was concluded to reflect the increased role of $O_3$ reactions in $H_2O_2$ formation, which is already tracked in our mechanism. The CO + OH reaction, producing a $\Delta^{17}O$ in the residual CO, would be extremely unlikely to affect the $\Delta^{17}O$

of $NO_y$ or $O_x$ due to the long atmospheric lifetime of CO relative to $NO_y$ or $O_x$. Therefore, we did not explicitly test these reactions' influence on $\Delta^{17}O$ of $NO_y$ or $O_x$ in this work but could easily be adapted in future iterations of the model.

The RACM2 mechanism is a gas-phase mechanism and does not include heterogeneous reactions, which could limit the ICOIN-RACM2 mechanism's ability to accurately simulate $\Delta^{17}O$ values, particularly of HONO and $HNO_3$ (Table 1). Gas-phase mechanisms are often used in larger chemical transport models that also include aerosol modules to calculate heterogeneous chemistry reaction rates. When utilizing ICOIN-RACM2 to simulate $\Delta^{17}O$ values (and RACM2 for simulating concentrations) in box models that lack aerosol modules, appropriate reactions should be included using pseudo-first-order reaction rate constants to calculate heterogeneous hydrolysis. However, estimating the heterogeneous reaction rates is not trivial and depends on the molecular speed, uptake coefficients, which depend on aerosol chemical composition, and surface area density. These reaction rates may need to be treated in a case-by-case circumstance. Since the ICOIN-RACM2 mechanism does not model particulate nitrate, we cannot model its photolysis, which could limit our ability to simulate $\Delta^{17}O$(HONO). Additionally, our gas-phase mechanism does not include $NO_2$ heterogeneous reactions, which could also be an important source of HONO (Chai et al., 2021). Users interested in accurately simulating $\Delta^{17}O$(HONO) may need to consider adding relevant reactions. Still, a future comparison between $\Delta^{17}O$(HONO) observations and model simulations based on the ICOIN-RACM2 framework shold provide pivotal insight into HONO formation.

## 2.2   Box Model Description

The ICOIN-RACM2 mechanism was utilized in the Framework for 0-D Atmospheric Modeling (F0AM) box-model (Wolfe et al., 2016). This box model presents a high degree of flexibility, allowing it to be seamlessly adapted for a wide range of simulation scenarios, and can perform online computation of photolysis frequencies. The ICOIN-RACM2 mechanism was developed for use in the F0AM. In this work, the F0AM model was utilized to illustrate the capacity of the ICOIN-RACM2 mechanism for simulating $\Delta^{17}O$ values from photochemical chamber experiments and steady-state diel cycles.

### 2.2.1   Chamber Simulations

Box model simulations were conducted to evaluate $\alpha$-pinene and $NO_x$ chemistry under various initial conditions that included variable [$\alpha$-pinene]:[$NO_x$] ratios (Table 3). These simulations were conducted using similar initial VOC and $H_2O_2$ levels utilized in recently conducted chamber experiments (Takeuchi and Ng, 2019). We have also varied the initial VOC and $NO_x$ concentration levels to look at the impact on changing initial conditions and model chemistry on $\Delta^{17}O$ values. $\alpha$-pinene is an important monoterpene, and its oxidation in the presence of $NO_x$ constitutes an important mechanism of coupled biogenic-anthropogenic interaction, with important consequences for air quality, climate, global reactive nitrogen budget, and secondary organic aerosols (SOA) (Romer et al., 2016; Zare et al., 2018; Ng et al., 2017). The model was initiated for each experiment using NO, $\alpha$-pinene, and $H_2O_2$ (OH precursor) concentrations. The $\alpha$-pinene and $H_2O_2$ concentrations were fixed at 25 ppb and 2,000 ppb, respectively, while the initial NO concentrations were varied from 5 to 125 ppb to simulate oxidation chemistry

**Table 2.** Summary of the considered O exchange reactions, and reaction rates in the ICOIN-RACM2 mechanism. These reactions were adapted from Lyons (2001).

| Label | Reaction | $k$ |
|---|---|---|
| O-Exchange01 | $Q(^3P) \xrightarrow{+O_2} O(^3P)$ | $2.9 \times 10^{-12}[O_2]$ $(s^{-1})$ |
| O-Exchange02 | $Q(^1D) \xrightarrow{+O_2} O(^1D)$ | $2.9 \times 10^{-12}[O_2]$ $(s^{-1})$ |
| O-Exchange03 | $Q(^1D) + NO \rightarrow O(^1D) + NQ$ | $3.7 \times 10^{-11}$ $(cm^3$ molecule$^{-1}$ $s^{-1})$ |
| O-Exchange04 | $O(^1D) + NQ \rightarrow Q(^1D) + NO$ | $3.7 \times 10^{-11}$ $(cm^3$ molecule$^{-1}$ $s^{-1})$ |
| O-Exchange05 | $Q(^3P) + NO \rightarrow O(^3P) + NQ$ | $3.7 \times 10^{-11}$ $(cm^3$ molecule$^{-1}$ $s^{-1})$ |
| O-Exchange06 | $O(^3P) + NQ \rightarrow Q(^3P) + NO$ | $3.7 \times 10^{-11}$ $(cm^3$ molecule$^{-1}$ $s^{-1})$ |
| O-Exchange07 | $QH \xrightarrow{+H_2O} OH$ | $2.3 \times 10^{-13}e^{(-2100/T(K))}[H_2O]$ $(s^{-1})$ |
| O-Exchange08 | $QH \xrightarrow{+O_2} OH$ | $1.0 \times 10^{-17}[O_2]$ $(s^{-1})$ |
| O-Exchange09 | $QH + HO_2 \rightarrow OH + HOQ$ | $1.0 \times 10^{-11}e^{(400/T(K))}$ $(cm^3$ molecule$^{-1}$ $s^{-1})$ |
| O-Exchange10 | $OH + HOQ \rightarrow 0.5QH + 0.5HO_2 + 0.5OH + 0.5HOQ$ | $1.0 \times 10^{-11}e^{(400/T(K))}$ $(cm^3$ molecule$^{-1}$ $s^{-1})$ |
| O-Exchange11 | $QH + HOQ \rightarrow 0.5OH + 0.5HQ_2 + 0.5QH + 0.5HOQ$ | $1.0 \times 10^{-11}e^{(400/T(K))}$ $(cm^3$ molecule$^{-1}$ $s^{-1})$ |
| O-Exchange12 | $OH + HQ_2 \rightarrow QH + HOQ$ | $1.0 \times 10^{-11}e^{(400/T(K))}$ $(cm^3$ molecule$^{-1}$ $s^{-1})$ |
| O-Exchange13 | $HOQ \xrightarrow{+O_2} HO_2$ | $3.0 \times 10^{-17}[O_2]$ $(s^{-1})$ |
| O-Exchange14 | $HQ_2 \xrightarrow{+O_2} HO_2$ | $3.0 \times 10^{-17}[O_2]$ $(s^{-1})$ |
| O-Exchange15 | $NQ + NO_2 \rightarrow NO + NOQ$ | $3.6 \times 10^{-14}$ $(cm^3$ molecule$^{-1}$ $s^{-1})$ |
| O-Exchange16 | $NO + NOQ \rightarrow 0.5NQ + 0.5NO_2 + 0.5NO + 0.5NOQ$ | $3.6 \times 10^{-14}$ $(cm^3$ molecule$^{-1}$ $s^{-1})$ |
| O-Exchange17 | $NQ + NOQ \rightarrow 0.5NO + 0.5NQ_2 + 0.5NQ + 0.5NOQ$ | $3.6 \times 10^{-14}$ $(cm^3$ molecule$^{-1}$ $s^{-1})$ |
| O-Exchange18 | $NO + NQ_2 \rightarrow NQ + NOQ$ | $3.6 \times 10^{-14}$ $(cm^3$ molecule$^{-1}$ $s^{-1})$ |
| O-Exchange19 | $NOQ \xrightarrow{+O_2} NO_2$ | $1.0 \times 10^{-24}[O_2]$ $(s^{-1})$ |
| O-Exchange20 | $NQ_2 \xrightarrow{+O_2} NO_2$ | $1.0 \times 10^{-24}[O_2]$ $(s^{-1})$ |
| O-Exchange21 | $QH + NO \rightarrow OH + NQ$ | $1.8 \times 10^{-11}$ $(cm^3$ molecule$^{-1}$ $s^{-1})$ |
| O-Exchange22 | $OH + NQ \rightarrow QH + NO$ | $1.8 \times 10^{-11}$ $(cm^3$ molecule$^{-1}$ $s^{-1})$ |
| O-Exchange23 | $QH + NO_2 \rightarrow OH + NOQ$ | $1.0 \times 10^{-11}$ $(cm^3$ molecule$^{-1}$ $s^{-1})$ |
| O-Exchange24 | $OH + NOQ \rightarrow 0.5QH + 0.5NO_2 + 0.5OH + 0.5NOQ$ | $1.0 \times 10^{-11}$ $(cm^3$ molecule$^{-1}$ $s^{-1})$ |
| O-Exchange25 | $QH + NOQ \rightarrow 0.5OH + 0.5NQ_2 + 0.5QH + 0.5NOQ$ | $1.0 \times 10^{-11}$ $(cm^3$ molecule$^{-1}$ $s^{-1})$ |
| O-Exchange26 | $OH + NQ_2 \rightarrow QH + NOQ$ | $1.0 \times 10^{-11}$ $(cm^3$ molecule$^{-1}$ $s^{-1})$ |

**Table 3.** Summary of the precursor concentrations for the box-model simulations of $\alpha$-pinene and NO photochemical oxidation chamber experiments. All experiments were simulated at a fixed temperature and relative humidity of 22 °C and 1 %, respectively.

| Exp. | $\alpha$-pinene (ppb) | $H_2O_2$ (ppb) | NO (ppb) | [$\alpha$-pinene]:[NO] |
|------|------------------------|----------------|----------|------------------------|
| 1 | 25 | 2,000 | 5 | 5:1 |
| 2 | 25 | 2,000 | 10 | 2.5:1 |
| 3 | 25 | 2,000 | 25 | 1:1 |
| 4 | 25 | 2,000 | 62.5 | 1:2.5 |
| 5 | 25 | 2,000 | 125 | 1:5 |

in a range of [$\alpha$-pinene]:[$NO_x$] conditions (Table 3). The pressure, temperature, and relative humidity were fixed at 1013 mbar, 295 K, and 1 %, respectively. The measured chamber light flux data from the Georgia Institute of Technology, Environmental Chamber Facility was also utilized.

The model was run for four hours for each simulated experiment. Both gas and particle chamber wall loss were not considered in the chamber simulation comparison. Monoterpene organic nitrate hydrolysis can be an important loss process and formation pathway of $HNO_3$ (Zare et al., 2018; Fisher et al., 2016; Takeuchi and Ng, 2019; Wang et al., 2021) but was not considered in the model because of the low relative humidity conditions. Additionally, heterogeneous pathways leading to the production of $HNO_3$, such as $N_2O_5$, were not included. For the hypothetical simulations, this should not impact the reliability of the predictions due to the photochemical conditions of the simulated chamber experiments, low relative humidity, and high organic carbon content of produced particles, which would be reasonably expected to lead to a low $N_2O_5$ uptake coefficient (Escorcia et al., 2010). When utilizing the ICOIN-RACM2 mechanism to simulate chamber experimental $\Delta^{17}O$ data, gas and particle wall-loss, organic nitrate hydrolysis, and $NO_y$ heterogeneous reactions should be considered, but it will depend on the chamber and reaction conditions and should be treated in a case-by-case circumstance. The model simulations evaluated the $\Delta^{17}O$ temporal variation of $NO_2$, HONO, monoterpene-derived organic nitrate (ONIT), $HNO_3$, OH, and $HO_2$ and investigated their changes in response to the experimental oxidant conditions.

## 2.2.2 Diel Variations

Box-model simulations were also conducted in steady-state diel cycles for two summertime scenarios. These scenarios (Case 19 and Case 20) were based on previous case studies utilized to evaluate the RACM and RACM2 mechanism (Stockwell et al., 1997; Goliff et al., 2013). Briefly, Case 19 represents a somewhat polluted atmosphere with emissions of $NO_x$ and organic compounds, and Case 20 represents a relatively cleaner atmosphere with the initial concentrations and emission rates of $NO_x$ and organic compounds reduced by a factor of 10 (Table 4). These scenarios would be analogous to near-surface summertime environments at an urban (i.e., Case 19) and rural (i.e., Case 20) environment. The box-model simulations were conducted for the initial conditions and with a fixed elevation of 0 km, temperature of 298 K, pressure of 1013.25 mbar, and for June 21

as previously described (Stockwell et al., 1997). The simulations were conducted for Providence, RI (41.82 °N, 71.41 °W), and the diel photolysis rates were calculated using the on-line module in F0AM (Wolfe et al., 2016). To avoid the buildup of

200 concentrations in the box model, a dilution lifetime of 24 hours was incorporated into the simulations, as previously described (Wolfe et al., 2016). The model simulations were run for five days at a one-hour interval. The first two days of the simulation were used as a spin-up period, and the diel cycles were evaluated based on the average of the final three days of the simulation.

In order to more accurately predict ambient atmosphere $\Delta^{17}O$ values, additional $NO_y$ heterogeneous reactions involving $NO_2$ and $N_2O_5$ were incorporated into the RACM2 mechanism and appropriately replicated for $Q$ isotopes in the ICOIN-RACM2

mechanism (Table 5). The reaction rate for $N_2O_5$ was taken from the Master Chemical Mechanism (MCM) version 3.3.1 estimate for a general ambient air scenario. The $NO_2$ heterogeneous reaction rate was then scaled based on the relative ratio of their uptake coefficients to $N_2O_5$ for an organic carbon aerosol with RH > 30 % (Holmes et al., 2019). While we do not claim these heterogeneous reaction rates are perfectly constrained nor could be generalized to other modeling scenarios, they are useful for the diel simulations to evaluate $\Delta^{17}O$ changes in daytime versus nighttime chemistry. The inclusion of

the $NO_2$ and $N_2O_5$ heterogeneous reactions in the utilized gas-phase mechanisms were termed RACM2(het) and ICOIN-RACM2(het), respectively. We note that the inclusion of organic nitrate hydrolysis was not considered in the diel simulations. This was because conditions for Case 19 and Case 20 did not include either an initial concentration or emission rate for BVOC compounds (Table 4). The diel simulations are used to demonstrate the utility of the ICOIN-RACM2 mechanism. The two near-surface summertime model scenarios do not represent all atmospheric conditions, including meteorology, actinic flux,

and emission rates, which will influence the model $\Delta^{17}O$ values. Thus, these simulations cannot quantitatively be compared with various field $\Delta^{17}O$ data. This type of comparison would require a more targeted simulation to represent the atmospheric conditions at a particular site, which is beyond the scope of this work. Still, we have compared qualitative trends predicted by the diel simulations with some available $\Delta^{17}O$ observations.

## 3 Results and Discussion

### 3.1 Mechanism Evaluation

The efficacy of the isotope tagging methodology was assessed through a comparative analysis of molecule concentrations using both the RACM2 and ICOIN-RACM2 mechanisms. For molecules encompassing oxygen isotopologues explicitly considered in the ICOIN-RACM2 mechanism, the concentrations were derived by summing the isotopologue concentrations (Eq. 4).

$$[X] = \sum_i [Z]_i \tag{4}$$

where $[X]$ refers to the concentration of a molecule with oxygen isotopologues, $i$ refers to the unique oxygen isotopologues, and $[Z_i]$ refers to the concentration of the $i$th isotopologue. Both the RACM2 and ICOIN-RACM2 mechanisms simulated identical concentrations across both simulated scenarios: the hypothetical chamber experiments and the diurnal variation case

**Table 4.** Conditions for the box-model simulations of the diel cycle for two summertime scenarios. The simulations were conducted at a fixed elevation of 0 km, temperature of 298 K, pressure of 1013.25 mbar, and on June 21, 2015, in Providence, RI (41.82 °N, 71.41 °W ). The scenarios were adapted from Stockwell et al. (1997).

| Compound | Case 19 | | Case 20 | |
| --- | --- | --- | --- | --- |
| | Initial Conc (ppb) | Emission Rate (ppt/hr) | Initial Conc (ppb) | Emission Rate (ppt/hr) |
| *Inorganics* | | | | |
| NO | 0.2 | 2.6 | 0.02 | 0.26 |
| $NO_2$ | 0.5 | — | 0.05 | — |
| $HNO_3$ | 0.1 | — | 0.01 | — |
| $O_3$ | 50 | — | 30 | — |
| $H_2O_2$ | 2.0 | — | 0.2 | — |
| $SO_2$ | — | 0.52 | — | 0.052 |
| CO | 200 | 5.7 | 104 | 0.57 |
| *Alkanes* | | | | |
| $CH_4$ | 1700 | — | 1700 | — |
| ETH | — | 0.24 | — | 0.024 |
| HC3 | — | 2.6 | — | 0.26 |
| HC5 | — | 0.76 | — | 0.076 |
| HC8 | — | 0.45 | — | 0.045 |
| *Alkenes* | | | | |
| ETE | — | 0.46 | — | 0.046 |
| OLI | — | 0.19 | — | 0.019 |
| OLT | — | 0.22 | — | 0.022 |
| *Aromatics* | | | | |
| TOL | — | 0.57 | — | 0.057 |
| XYL | — | 0.52 | — | 0.052 |
| *Carbonyls* | | | | |
| HCHO | 1.0 | 0.14 | 0.1 | 0.014 |
| ALD | — | 0.036 | — | 0.0036 |
| KET | — | 0.32 | — | 0.032 |

**Table 5.** Summary of the $NO_y$ heterogeneous reactions, assumed uptake coefficients, and the reference or calculated pseudo-first order reaction rates adapted in the RACM2(het) and ICOIN-RACM2(het) chemical mechanisms.

| Label | Reaction | $\gamma^a$ | $k_{het}(s^{-1})$ |
|---|---|---|---|
| Het01 | $NO_2 \rightarrow 0.5HNO_3 + 0.5HONO$ | $(10^{-6})$ | $(2.67 \times 10^{-6})^b$ |
| Het02 | $N_2O_5 \rightarrow 2HNO_3$ | $(1.5 \times 10^{-4})$ | $(4.0 \times 10^{-4})^c$ |

[a] Adapted from Holmes et al. (2019)

[b] Calculated by scaling the $k_{het}$ ($N_2O_5$) based on the relative $\gamma$

[c] Taken from the MCM v3.3.1 for general ambient scenarios.

study during the summer period (Fig. 1). This congruence in results aligns with expectations, as the isotope tagging approach implemented in the ICOIN-RACM2 is designed not to alter the chemical kinetics governing gas-phase reactions. Indeed, by
definition, the presence of isotopes should remain inert with regard to chemical reactivity. This comparative analysis serves as a robust validation of the isotope tagging methodology's ability in simulating $\Delta^{17}O$ values while maintaining the chemical reactivity stipulated by the RACM2 mechanism.

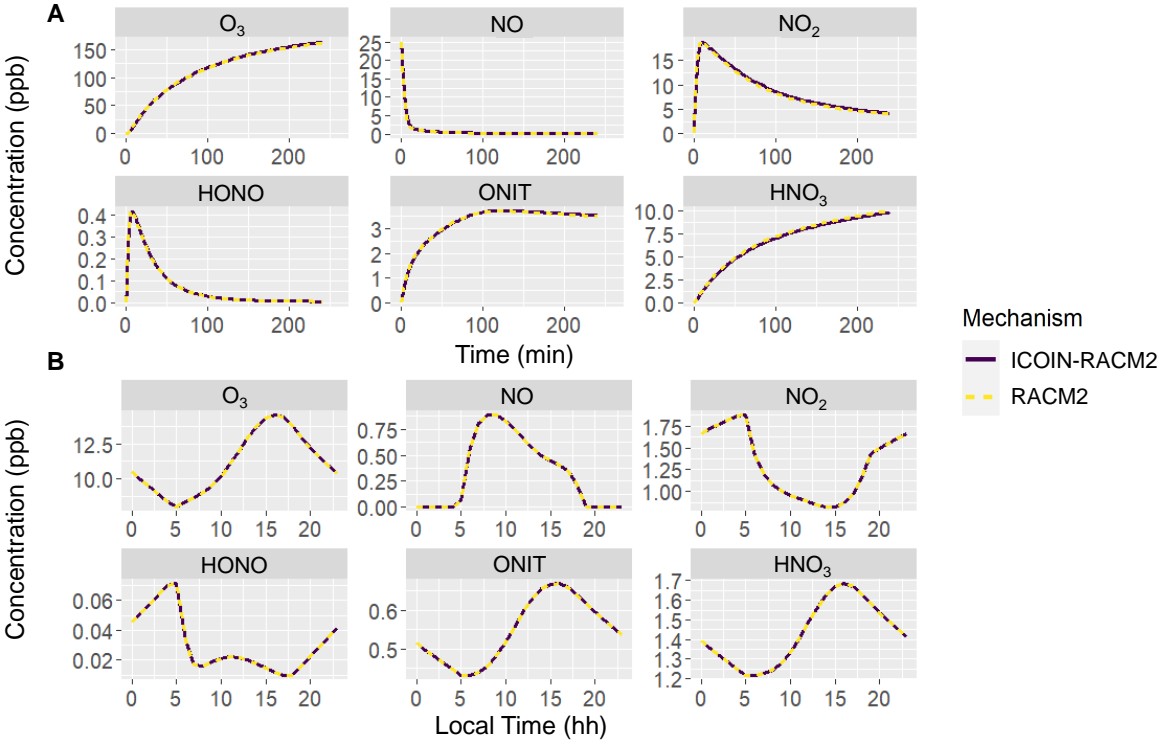

**Figure 1.** Comparison of the RACM2 (dashed yellow line) and ICOIN-RACM2 (purple solid line) mechanisms for simulating concentrations of several molecules for (A) Chamber Experiment-3 and (B) Diel Cycle-Case 19.

## 3.2  Chamber Simulations of $NO_x/\alpha$-pinene Chemistry

The simulated $\Delta^{17}O$ values derived from the hypothetical chamber simulations reveal significant temporal variations (Fig. 2). Overall, there were significant differences in $\Delta^{17}O$ values across the considered molecules that increased in the sequence of $HO_2$, OH, $HNO_3$, HONO, and $NO \approx NO_2$. In the photochemical model simulations, the $\Delta^{17}O(NO) \approx \Delta^{17}O(NO_2)$ due to both rapid $NO_x$ photochemical cycling as well as $NO/NO_2$ isotope exchange. Among the considered $NO_y$ molecules, the initial $\Delta^{17}O$ values start at 0 ‰ and subsequently rises due to the generation of $O_3$ and subsequent propagation into the $NO_y$ components, which leads to heightened $\Delta^{17}O$ values. The extent of $\Delta^{17}O$ elevation was determined to be contingent upon the initial chamber conditions, becoming more pronounced with increasing initial NO to $\alpha$-pinene ratios for $\Delta^{17}O$ of NO, $NO_2$, HONO, and $HNO_3$. In contrast, $\Delta^{17}O(HO_2)$ was nearly negligible, aligning with common assumptions in other $\Delta(^{17}O)$ models (Alexander et al., 2020, 2009; Michalski et al., 2003; Morin et al., 2011). Similarly, $\Delta^{17}O(OH)$ generally maintained close proximity to 0 ‰, in line with typical assumptions in other $\Delta(^{17}O)$ models (Alexander et al., 2020, 2009; Michalski et al., 2003; Morin et al., 2011); although there were instances that deviated from this trend. Notably, higher $\Delta^{17}O(OH)$ values were observed as the initial $NO_x$ relative to BVOC concentrations increased. This occurrence can be attributed to the increased significance of oxygen isotope exchange between $NO_2$ and OH for the higher initial $NO_x$ experimental conditions.

An intriguing observation was that the simulated $\Delta^{17}O(ONIT)$ values remained unaffected by the chamber's initial conditions (Fig. 3). This observation underscores the diverse ONIT formation pathways present in the experiments, encompassing a $\Delta^{17}O(ONIT)$ low-end pathway involving $\alpha$-pinene peroxy radical (APIP; a type of $RO_2$) + NO, and a high-end pathway involving nitrooxy peroxy ($nRO_2$) deriving from $\alpha$-pinene + $NO_3$ reactions (Table 1). We note that even though all of the simulated experiments were conducted under photochemical conditions, the model predicted some oxidation of $\alpha$-pinene with $NO_3$. The relative proportion of these two significant ONIT formation routes exhibited substantial variability across the various experiments (Fig. 3). Generally, a higher fractional formation of ONIT occurred through the $\Delta^{17}O$ high-end member pathway of $nRO_2$ + Y (where Y = $HO_2$, NO, $nRO_2$, $ACO_3$, $MO_2$) as the initial NO to $\alpha$-pinene ratios were lower. Additionally, we note that the produced $\Delta^{17}O(ONIT)$ value is a balance between the ONIT production pathway and the $\Delta^{17}O(NO)$ (Table 1). For the lower initial $[NO_x]:[\alpha$-pinene] experiments, a lower $\Delta^{17}O(NO)$ value was simulated. The balance between $\Delta^{17}O(NO)$ and the pathway leading to ONIT production can explain the observation that $\Delta^{17}O(ONIT)$ was insensitive to initial conditions.

The simulated $\Delta^{17}O$ values from the chamber experiments highlight compelling dynamics. Primarily, there were substantial differences in $\Delta^{17}O$ values arising from different formation pathways contributing to ONIT production. The uncertain nature of branching ratios and product yields for ONIT underscores the potential utility of comparing $\Delta^{17}O(ONIT)$ observations with model simulations, aiding in the refinement of our understanding of ONIT yields originating from APIP + NO and API + $NO_3$ reaction pathways. Furthermore, a significant divergence between $HNO_3$ and ONIT in terms of $\Delta^{17}O$ was observed, particularly with heightened initial NO to $\alpha$-pinene concentrations. This divergence led to considerably higher simulated $\Delta^{17}O(HNO_3)$ than $\Delta^{17}O(ONIT)$. This difference could potentially be utilized to help constrain the contribution of ONIT hydrolysis to $HNO_3$ through a comparison of observed $\Delta^{17}O(HNO_3)$ and model-based predictions. Lastly, the model simulations underscore the

potential of employing the ICOIN-RACM2 model to corroborate $\Delta^{17}O$ values for OH and $HO_2$ under diverse conditions (such as concentrations, chemical composition, relative humidity, and temperature), which are commonly assumed to be near 0 ‰.

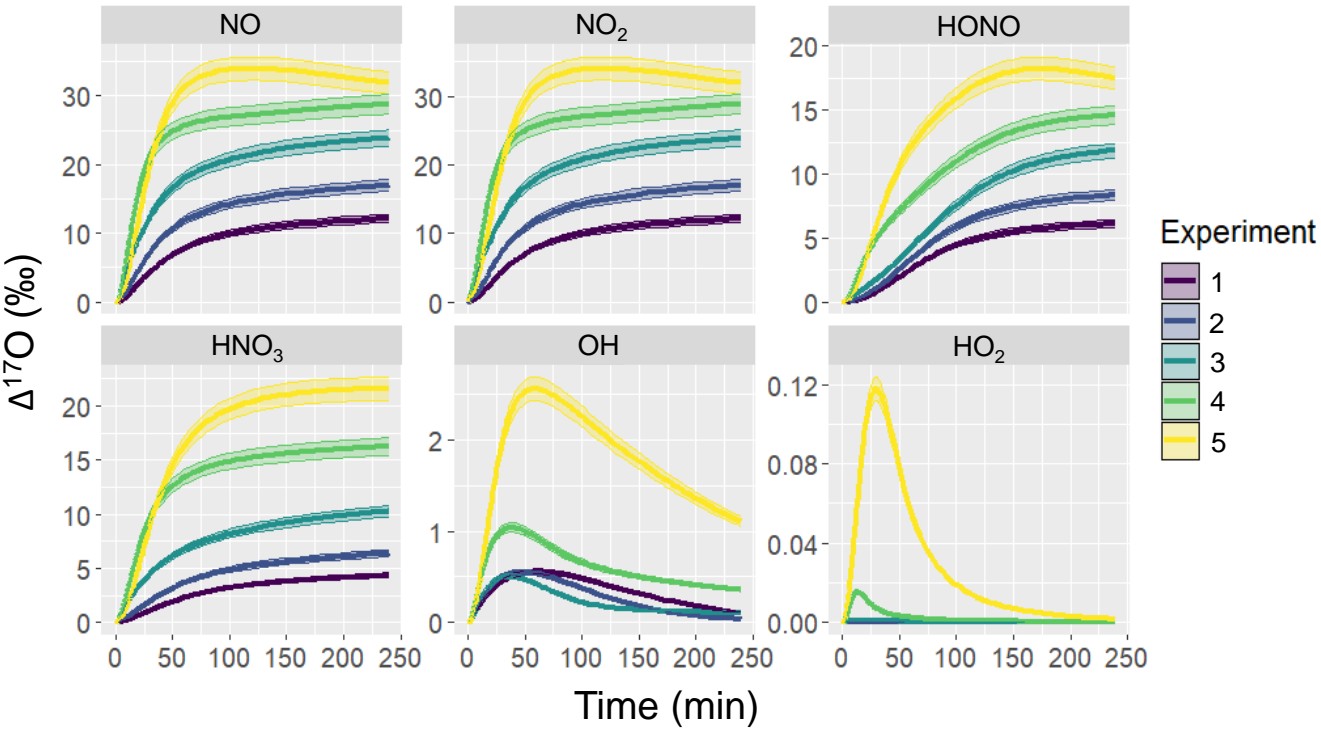

**Figure 2.** Simulation of $\Delta^{17}O$ for several $NO_y$ and $O_x$ molecules including, NO, $NO_2$, HONO, $HNO_3$, OH, and $HO_2$ for the various hypothetical chamber experiments (color-coded). The solid line represents the modeled $\Delta^{17}O$ value, and the shaded region corresponds to the propagated uncertainty associated with the chosen $\Delta^{17}O(O_3{}^{term})$ value of 39.3 ±2 ‰. The experimental initial conditions are provided in Table 3, which include a starting NO concentration of 5, 12.5, 25, 62.5, and 125 ppb for experiments 1, 2, 3, 4, and 5, respectively.

### 3.3 Summertime Diel Simulations

The simulated $\Delta^{17}O$ diel profiles indicate interesting patterns for the various considered molecules, including NO, $NO_2$, HONO, $HNO_3$, OH, and $HO_2$ (Fig. 4). The $\Delta^{17}O$ of NO, $NO_2$, and HONO indicates a strong diurnal pattern for both considered summertime case studies. The simulated $\Delta^{17}O$ of NO and $NO_2$ indicates lower values during the nighttime and higher values during the daytime. This is due to the importance of nighttime NO emissions, such that the O atoms of NO and $NO_2$ are not photochemically cycled. The simulated daytime profiles of $\Delta^{17}O$ of NO and $NO_2$ follow similar patterns, reflecting their fast photochemical cycling. Near sunrise, $\Delta^{17}O$ of NO and $NO_2$ reaches a peak due to photochemical cycling that primarily involves $O_3$. As photolysis continues, there is a significant enhancement of peroxy radicals ($RO_2/HO_2$), which slightly dilutes the $\Delta^{17}O$ of NO and $NO_2$. Near sunset, the peroxy radical concentrations decrease, and once again, NO and $NO_2$ pre-

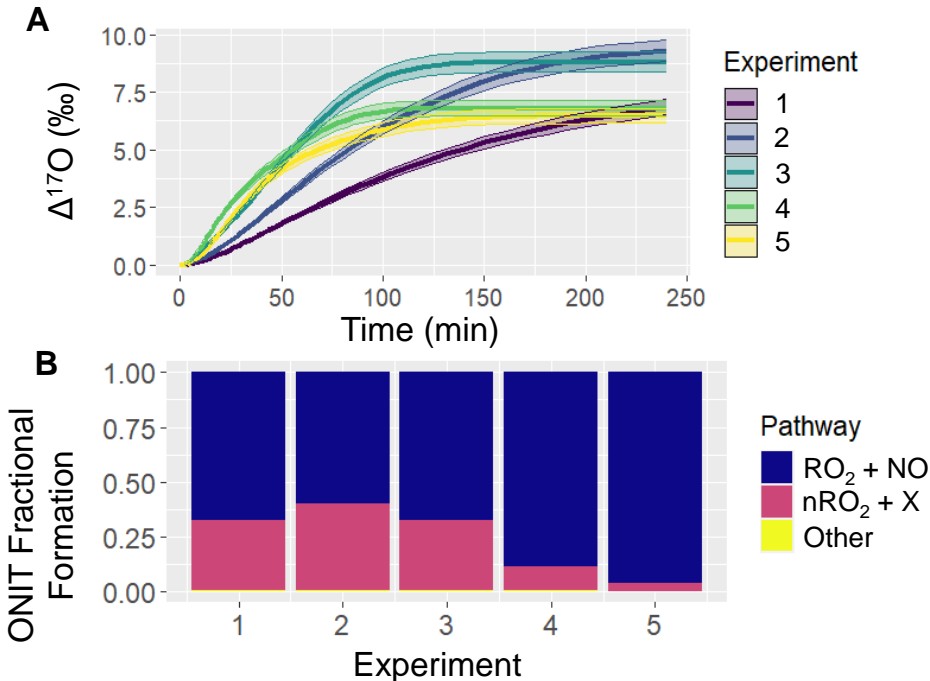

**Figure 3.** Simulation of ONIT chemistry for the various considered chamber experiments including: (A) $\Delta^{17}O$ of ONIT and (B) ONIT fractional formation pathways. The solid line represents the modeled $\Delta^{17}O$ value, and the shaded region corresponds to the propagated uncertainty associated with the chosen $\Delta^{17}O(O_3{}^{term})$ value of 39.3 ±2 ‰ (A). The experimental initial conditions are provided in Table 3, which include a starting NO concentration of 5, 12.5, 25, 62.5, and 125 ppb for experiments 1, 2, 3, 4, and 5, respectively.

dominantly photochemically cycle with $O_3$. During the daytime, the simulated $\Delta^{17}O(NO_2) \approx \Delta^{17}O(NO)$, due to due to the rapid $NO_x$ photochemical cycling. However, during the nighttime, $\Delta^{17}O(NO_2)$ was greater than $\Delta^{17}O(NO)$ due to the role of nighttime NO emissions with an assumed $\Delta^{17}O(NO) = 0$ ‰. While NO and $NO_2$ isotope exchange would lead to $\Delta^{17}O(NO)$

$= \Delta^{17}O(NO_2)$, its role in influencing $\Delta^{17}O$ depends on the concentrations of NO and $NO_2$, as previously discussed for $\delta^{15}N$ of $NO_x$ (Walters et al., 2016). In the diel model simulations, nighttime NO concentrations were less than 0.1 ppb (Fig. 1) due to its titration by $O_3$. Under these conditions, the rate of NO/$NO_2$ isotope exchange was slow relative to NO oxidation or the rate of NO primary emission, leading to a low nighttime $\Delta^{17}O(NO)$ value for the simulation conditions of low nighttime $NO_x$ relative to $O_3$ concentrations.

The predicted $NO_2$ diurnal cycles of elevated $\Delta^{17}O$ during the daytime and low $\Delta^{17}O$ during the nights are generally consistent with summertime $\delta^{18}O$ observations (which track with $\Delta^{17}O$) in West Lafayette, IN, US (Walters et al., 2018), and recent diel observations of $\Delta^{17}O$ at Grenoble, FR, during the spring (Albertin et al., 2021). However, there are some slight differences in the daytime $\Delta^{17}O(NO_2)$ observations compared to the model simulations, in which the highest $\Delta^{17}O(NO_2)$ occurred for sam-

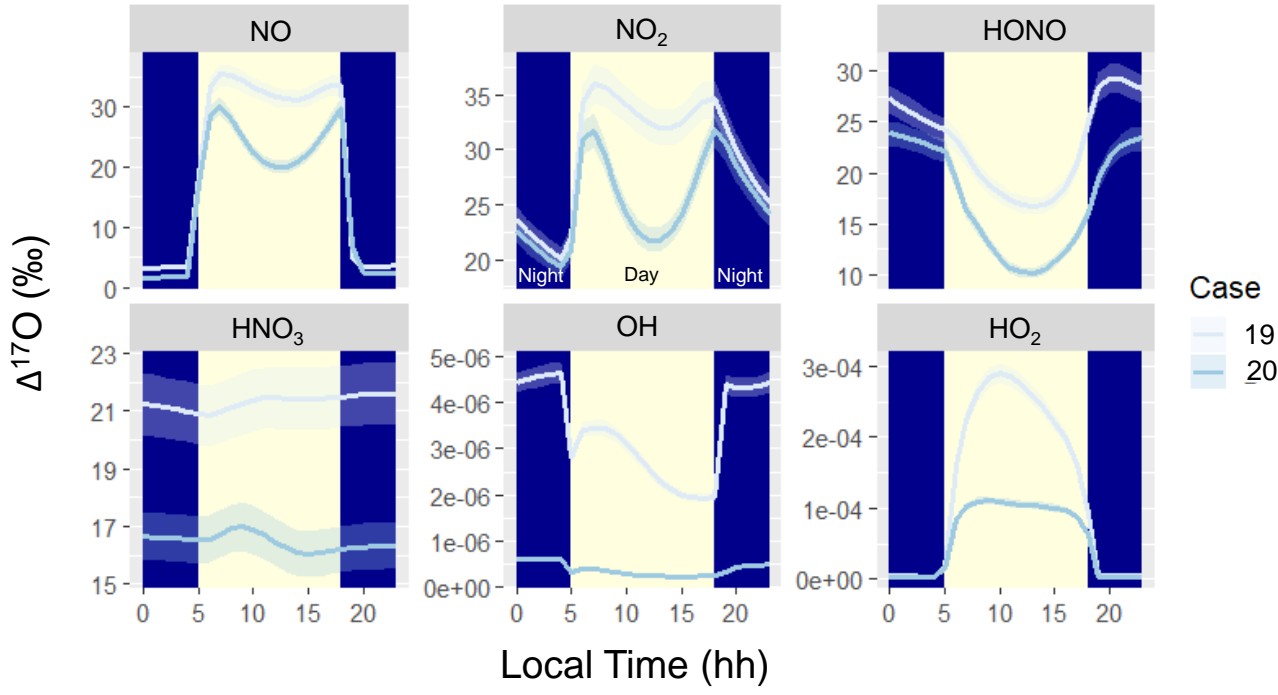

**Figure 4.** Diel simulation of $\Delta^{17}O$ for several $NO_y$ and $O_x$ molecules including, NO, $NO_2$, HONO, $HNO_3$, OH, and $HO_2$ for the various hypothetical summertime cases (color-coded). The solid line represents the modeled $\Delta^{17}O$ value, and the shaded region corresponds to the propagated uncertainty associated with the chosen $\Delta^{17}O(O_3{}^{term})$ value of 39.3±2‰. The shading corresponds to daytime (light yellow) and nighttime (dark blue) conditions.

ples collected between 9 am – 12 pm (Albertin et al., 2021). In comparison, the model indicated the highest $\Delta^{17}O(NO_2)$ around 6 to 8 am following the return of photolysis near sunrise. The observations indicate a subsequent daytime decay of $\Delta^{17}O(NO_2)$ (Albertin et al., 2021). The model also indicates a daytime decay in $\Delta^{17}O(NO_2)$ following the initial maximum $\Delta^{17}O(NO_2)$ that coincides with the onset of photolysis; however, the model expects an increase in $\Delta^{17}O(NO_2)$ in the late afternoon due to increased $O_3/HO_x$ levels from the decrease in actinic flux. We do not intend to accurately simulate the previously reported $\Delta^{17}O(NO_2)$ values (Albertin et al., 2021). Some of the nuanced differences between the model simulation and observations of $\Delta^{17}O$ are likely due to differences in meteorological conditions, as the model was simulated for summertime while the observations were from springtime and for a different latitude and longitude. Further, our model neglects transport and assumes a constant emission rate, which could influence the diel $\Delta^{17}O(NO_2)$ predictions. Nevertheless, the ICOIN-RACM2 mechanism appears to capture the general diurnal trend of $\Delta^{17}O(NO_2)$. We envision that future adaptation of the ICOIN-RACM2 mechanism into a chemical transport model would provide useful insight for constraining $NO_x$ photochemical cycling based on a comparison to field $\Delta^{17}O(NO_2)$ measurements.

The diurnal variation in $\Delta^{17}O(HONO)$ exhibits an inverse pattern compared to NO and $NO_2$, characterized by nocturnal maxima and daytime minima. This contrast arises from distinct formation pathways operating during daytime and nighttime. For our model simulations and conditions, HONO formation predominantly occurs via $NO_2$ heterogeneous reactions during the night, giving rise to a high-$\Delta^{17}O(HONO)$ end-member (Table 1). Conversely, daytime HONO production centers around the NO + OH pathway, leading to a low-$\Delta^{17}O(HONO)$ end-member, which dilutes the $\Delta^{17}O$ of the formed HONO relative to $\Delta^{17}O$ of HONO and NO (Table 1). Notably, primary emissions could significantly contribute to HONO levels but were excluded from the hypothetical summertime scenarios (Stockwell et al., 1997). If primary HONO emissions were substantial, a lower $\Delta^{17}O(HONO)$ during the night would be anticipated due to a lack of $NO_y$ photochemical cycling, assuming primary emissions carry a $\Delta^{17}O(HONO)$ value of 0 ‰. Additionally, we note that the model, based on a gas-phase mechanism, does not include photolysis of $pNO_3$, which could be an important source of HONO (Ye et al., 2016). For future interpretation of $\Delta^{17}O(HONO)$ observations, photolysis of $pNO_3$ as well as optimized $NO_2$ heterogeneous reaction rates would need to be considered.

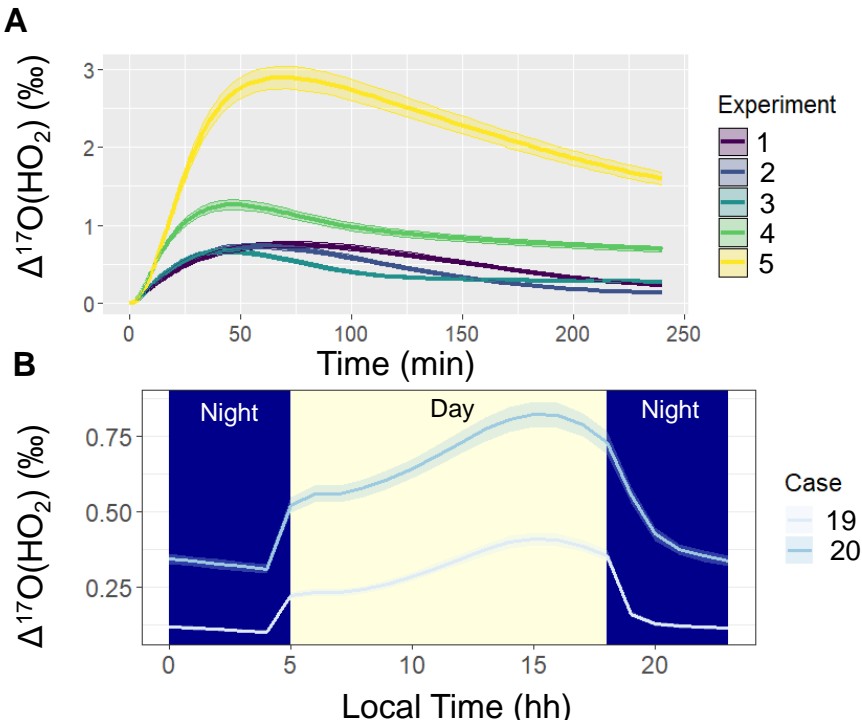

**Figure 5.** Simulated $\Delta^{17}O(HO_2)$ when not considering oxygen isotope exchange between $HO_2$ and $O_2$ in the ICOIN-RACM2 mechanism (i.e., O-Exchange13 and O-Exchange14 in Table 2) for (A) the considered chamber simulations (color-coded) and (B) the diel simulations for hypothetical summertime cases (color-coded). The solid line represents the modeled $\Delta^{17}O$ value, and the shaded region corresponds to the propagated uncertainty associated with the chosen $\Delta^{17}O(O_3{}^{term})$ value of 39.3±2 ‰. The shading corresponds to daytime (light yellow) and nighttime (dark blue) conditions for the diel simualtions (B).

The $\Delta^{17}O$ of $HNO_3$, OH, and $HO_2$ had little variation in their diel profiles. The $\Delta^{17}O(HNO_3)$ tended to converge to a value dependent on the oxidant conditions for Case 19 and Case 20. There were no significant simulated $\Delta^{17}O(HNO_3)$ diurnal vari-

ability due to the relatively longer $HNO_3$ lifetime in the gas-phases mechanism relative to NO, $NO_2$, and HONO, which rapidly undergo photochemical cycling. In the RACM2 mechanism, the significant chemical loss pathways for $HNO_3$ are $HNO_3$ + OH and $HNO_3$ photolysis, which are relatively slow loss pathways, essentially "locking-in" the $\Delta^{17}O(HNO_3)$ values. Thus, due to the relatively elevated $HNO_3$ lifetime, the simulated $\Delta^{17}O(HNO_3)$ builds up toward a steady-state value. While the modeled diel $\Delta^{17}O(HNO_3)$ indicated no substantial diurnal variations, several field studies have indicated significant diurnal variability

of $\Delta^{17}O(pNO_3)$ in polluted mega-cities (Zhang et al., 2022), as well as off the coast of California (Vicars et al., 2013). Commonly, $\Delta^{17}O(HNO_3)$ is thought to be equal to $\Delta^{17}O(pNO_3)$ due to the thermodynamic equilibrium between $HNO_3$ and $pNO_3$ in the fine aerosol mode (Alexander et al., 2009). However, recent data would suggest that $\Delta^{17}O(HNO_3)$ may not always be equal to $\Delta^{17}O(pNO_3)$ due to contributions of $pNO_3$ in the coarse aerosol phase that may not achieve thermodynamic equilibrium with $HNO_3$ (Kim et al., 2023). If we consider that the $\Delta^{17}O(pNO_3)$ diurnal variability should follow $\Delta^{17}O(HNO_3)$, the

discrepancy between model and observations of diurnal variability would suggest that the lifetime of $pNO_3$ in these previous studies must be shorter than predicted in our model for $HNO_3$. Our model simulation was conducted using a gas-phase mechanism within a simple box-model framework. Potentially important $pNO_3$ loss processes not included in our model include $pNO_3$ photolysis and wet/dry deposition. These processes should not alter the $\Delta^{17}O$ of $pNO_3$ but could reduce the lifetime of $pNO_3$, leading to a significant diurnal variation in $\Delta^{17}O$. Additionally, our model simulation does not include transport

or changes in boundary layer height and break up of the nocturnal boundary layer, which could also influence $\Delta^{17}O$ diurnal variations of $HNO_3$ and $pNO_3$.

The simulated $\Delta^{17}O$ of OH and $HO_2$ was near 0 ‰ for the entire simulation. We note that the simulated $\Delta^{17}O(HO_2)$ was lower than previous $\Delta^{17}O(HO_2)$ simulations (Morin et al., 2011), which tended to be between 1 to 2 ‰. This difference is because we have included oxygen isotope exchange reactions involving $O_2$ and $HO_2$ (Lyons, 2001) (i.e., O-Exchange13 and O-Exchange14

in Table 2) in the ICOIN-RACM2 mechanism, which rapidly remove $\Delta^{17}O > 0$ ‰ in the generated $HO_2$. Without including this oxygen isotope exchange reaction, the ICOIN-RACM2 modeled $\Delta^{17}O(HO_2)$ predicts a non-zero $\Delta^{17}O(HO_2)$ that can be as high as 3 ‰ dependent on the model conditions (Fig. 5), consistent with previous model simulations (Morin et al., 2011). While the $\Delta^{17}O(HO_2)$ is expected to have a minor impact on the $\Delta^{17}O$ of $NO_y$ species (Alexander et al., 2009), we should consider the importance of the role of oxygen isotope exchange between $O_2$ and $HO_2$ influencing $\Delta^{17}O(HO_2)$, as it will be an

important source of $\Delta^{17}O$ of $H_2O_2$, which is propagated into atmospheric sulfate (Savarino et al., 2000).

Comparing Case 19 and Case 20 reveals coherent diel $\Delta^{17}O$ patterns. The primary disparity between these case studies lies in the higher $\Delta^{17}O$ values exhibited by the considered $NO_y$ compounds in the urban setting of Case 19 compared to the rural backdrop of Case 20 (Fig. 4). The general divergence between the urban and rural simulated $\Delta^{17}O$ stems from the interplay between $O_3$ and $RO_2/HO_2$. Urban conditions (i.e., Case 19) entail greater contributions from $NO_x$ photochemical cycling with

$O_3$ relative to the rural environment (i.e., Case 20). An important exception to the urban vs rural trend was for $\Delta^{17}O(HO_2)$ when $O_2/HO_2$ exchange reactions were not considered in the model simulation (Fig. 5). From these simulations, the rural $\Delta^{17}O(HO_2)$

was slightly larger than the urban simulations, reflecting differences in the involvement of $O_3$ in $HO_2$ formation between these simulations. Overall, these hypothetical simulations highlight the prospect of intriguing $\Delta^{17}O$ variations between urban and rural settings.

## 4 Conclusions

This study introduces a novel gas-phase mechanism, denoted as ICOIN-RACM2, which is built upon the RACM2 gas-phase chemical mechanism framework (Goliff et al., 2013). This mechanism is designed to explicitly model the $\Delta^{17}O$ of $NO_y$ and $O_x$ molecules based on quantitatively tracking the incorporation and propagation of the oxygen isotope anomaly derived from $O_3$. Its application is demonstrated through box-model simulations encompassing diverse hypothetical scenarios. These scenarios include chamber experiments focused on $\alpha$-pinene and NO photochemical oxidation, as well as the exploration of diurnal cycles in summertime chemistry.

These initial investigations serve as a fundamental step towards advancing our comprehension of $NO_y$ oxidation chemistry and its intricate pathways. Notably, the mechanism exhibits promising capabilities in simulating $\Delta^{17}O$ values for multiple $NO_y$ and $O_x$ species. This capacity holds considerable promise for refining our insights into aspects such as ONIT formation, branching ratios, and hydrolysis dynamics. Moreover, the ICOIN-RACM2 mechanism emerges as a valuable tool for various air-quality-related objectives. It could be an invaluable tool for the assessment of primary emission strengths for HONO and the probing of urban-to-rural gradients of atmospheric oxidation chemistry. In forthcoming endeavors, this newly devised model will be instrumental in direct comparisons with $\Delta^{17}O$ observations arising from chamber experiments and investigations tied to air quality. As techniques for analyzing $\Delta^{17}O$ in $NO_y$ molecules continue to advance, the model's utility is poised to expand.

An envisioned next step involves integrating the model into a broader 3-D atmospheric chemistry framework. This integration is anticipated to offer vital insights for evaluating the representation of oxidation chemistry across diverse landscapes. These endeavors have far-reaching implications, notably in fine-tuning our capacity to accurately model and predict atmospheric oxidation processes, thus enhancing our overall understanding of atmospheric oxidation capacity.

*Code availability.* The developed mechanism, box-model source codes, and the input and output files have been made publicly available at https://zenodo.org/records/10961373 (doi: 10.5281/zenodo.10961373).

*Author contributions.* WWW designed, tested, and evaluate the newly developed mechanism. MT and NLN provided critical insight into utilizing box-models to simulate chamber experimental data. WWW and MGH secured funding for this work. WWW prepared the article with contributions from all co-authors.

*Competing interests.* The contact author has declared that none of the authors have any competing interests.

*Financial support.* This research has been supported by NOAA's Climate Program Office's Atmospheric Chemistry, Carbon Cycle, and Climate program (NOAA AC4 NA18OAR4310118).

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
