# Peer review of "Incorporating Oxygen Isotopes of Oxidized Reactive Nitrogen in the Regional Atmospheric Chemistry Mechanism, Version 2 (ICOIN-RACM2)"

_EGUsphere, 2023_

## Referee Comment (RC1)

**Comments on manuscript egusphere-2023-2293, "Incorporating Oxygen Isotopes of Oxidized Reactive Nitrogen in the Regional Atmospheric Chemistry Mechanism, Version 2 (ICOIN-RACM2)", by Wendell W. Walters et al.**

In this paper, Walters et al. presented an isotopic chemical mechanism (ICOIN-RACM2) by incorporating oxygen isotopes into the gas-phase chemical mechanism RACM2. This new isotopic mechanism aims to simulate the oxygen mass-independent fractionation signals ($\Delta^{17}O$) of atmospheric oxidized reactive nitrogen ($NO_y$) and odd oxygen species. The authors adopted an isotope-tagging method by explicitly tracking the oxygen atom that was transferred from the terminal of ozone in each molecule. After examining the consistency between the new isotopic mechanism and the original RACM2 mechanism, the authors conducted two case studies by applying 0-D box model simulations. The first is to simulate the $\Delta^{17}O$ of NOy in ideal photochemical chamber experiments with different initial conditions, and the second is to simulate the diel cycles of $\Delta^{17}O$ in a summertime atmosphere. The author concluded that this new mechanism would help advance our understanding of $NO_y$ oxidation chemistry and is expected to be useful for future applications in 3-D CTMs.

Oxygen isotope signals, especially $\Delta^{17}O$, have been widely used to constrain the oxidation chemistry of atmospheric reactive nitrogen. The novel isotopic tagging chemical mechanism introduced by this study could serve as a useful tool in such studies. I feel the paper is well written and the topic is of great interest for the readership of GMD. However, I have three major concerns on this study:

(1) The assumption of a constant $\Delta^{17}O$ of ozone. The author cited the study of Vicars and Savarino et al. (2014) to validate the choice of a constant $\Delta^{17}O(O_3)$. I agree that the **average** tropospheric $\Delta^{17}O(O_3)$ from multiple observational studies is close to 26 ‰, but there are also significant seasonal cycles in these observations, including the Vicars and Savarino. (2014) study. For example, Savarino et al. (2016) suggested an ~10 ‰ variation in $\Delta^{17}O(O_3)$ in different seasons at Dome C. Xu et al. (2022) also indicated a >5 ‰ variation in the $\Delta^{17}O$ of the ozone terminal oxygen in Japan (Xu et al., 2022), which was attributed to a stratospheric intrusion event. These values are already large enough to impact the oxygen MIF transferred to other molecules from ozone. Moreover, it is well known that $\Delta^{17}O(O_3)$ is sensitive to both temperature and pressure (Thiemens and Jackson, 1990; Janssen et al., 2003) and that stratospheric ozone has a much greater $\Delta^{17}O$ than tropospheric ozone (Krankowsky et al., 2007). Therefore, adapting an average value from mostly near-surface observations is a major drawback of the method used in this study and severely limits the extension of the proposed mechanism to 3D CTMs, as the author suggested.

(2) The ignorance of other processes that could produce OMIF. There are several known chemical processes that can produce extra OMIF, e.g., the CO+OH reaction and the $HO_2+HO_2$ reaction (Röckmann et al., 1998; Velivetskaya et al., 2016), and perhaps additional unknown reactions exist. From my point of view, the isotopic tagging method can hardly handle these effects. The impact of these

processes on the $\Delta^{17}O$ of atmospheric NOy could be small but should not be ignored in this study.

(3) Comparisons with observations to validate the proposed mechanism are lacking. Although the author conducted two different case simulations in this study, it is unclear why these cases were chosen. For the chamber simulations, it looks that the model configurations mimic the chamber experiments in Blum et al. (2023), but why the author didn't compare the model results with Blem et al. (2023)? This would provide the reader with a direct impression about the model performance. I noticed that in Blem et al. (2023), both $\Delta^{17}O(NO_2)$ and $\Delta^{17}O(NO_3^-)$ were measured, and the comparison should be straightforward. In addition, the measured $\Delta^{17}O(NO_2)$ in Blem et al. (2023) could exceed 40 ‰, which cannot be reproduced by assuming a constant 39.3 ‰ $\Delta^{17}O$ of terminal oxygen in ozone in this study. The modeled atmospheric $\Delta^{17}O(NO_2)$ trend in the latter case was also not consistent with recent observations (see my minor comment below). These simple comparisons suggested the limitations or incompetence of the current mechanism.

Nevertheless, I appreciate much about the effort made by this study, and I look forward to incorporating this new oxygen isotopic mechanism into 3D CTMs in the future. I suggest the author to include direct comparisons with the existing observational data especially the chamber experiments, which would require a redesign of the box model simulations, and the author should make more explanations on the points I mentioned above in the revised manuscript. For these reasons I think a major revision is necessary before the final publication of this manuscript on GMD.

**- Minor comments**

- I don't understand why the author used to added a parenthesis in the delta notation throughout the text. I suggest the author to follow the most conventional notation by using $\delta^{18}O$ and $\Delta^{17}O$.
- Lines 47-48: Walters et al., 2018, Blum et al., 2020 and Chai and Hastings, 2018 didn't measure $\Delta^{17}O$ in their study and are improper citations here.
- Line 59-62: I suggest to rephrase this sentence. Clearly, photochemical equilibrium is no longer held at night because there is no light or photolysis of $NO_2$, instead of nocturnal NO emission. I understand the author may want to describe that the freshly emitted NO would dilute the residual $\Delta^{17}O(NO_2)$ from the daytime, but the statement seems weird to me.
- Table 1: The $\Delta^{17}O(RO_2NO_2)$ should be $1/2*\Delta^{17}O(RO_2)$ plus $1/2*\Delta^{17}O(RO_2)$.
- Section 2.1: Does the model incorporate the photolysis of particle nitrate? The author mentioned the importance of this process but it's not clear if it is included by the model. In addition, I suggest the author to add a table that describing the reaction mechanism in a supplemental file, otherwise the reader not familiar with the RACM2 mechanism would have to search the model code, which is a rather tough task.
- Figure 1 and Figure 2: I suggest to change the y scales in different subplots.

- Line 201-202: The simulated $\Delta^{17}O$ of NO was described in this sentence but not shown in the figure. Please add it
- Line 297-298: Another possible explanation for the nonzero $\Delta^{17}O$ of NO is the isotopic exchange between NO and $NO_2$. The author should easily test this hypothesis by turning off the $NO-NO_2$ exchange reaction in the mechanism.
- Line 298-234: While the overall day high and night low trend in $\Delta^{17}O(NO_2)$ was reproduced by the model, the simulated **diurnal** $\Delta^{17}O(NO_2)$ trend was indeed **not consistent** with the observations in Albertin et al. (2021). Albertin et al. (2021) reported that the maximum $\Delta^{17}O(NO_2)$ occurred at noon, while the model suggested a minimum at noon. New observations by Albertin et al. (2024) indicated a similar trend. This point deserves further discussion. Moreover, in Figure 4, the simulated diurnal $\Delta^{17}O(NO_2)$ seems to be slightly greater than $\Delta^{17}O(NO)$, especially in case 20. This is astonishing since both photochemical cycling and isotopic exchanges tend to force them equal. Please explain it.
- Line 235: The discussion of the diurnal trend in $\Delta^{17}O(HONO)$ needs more attention because the photolysis of $p-NO_3^-$ seems to not be included by the model, which is however crucial to the HONO budget in many environments (e.g., (Ye et al., 2016)). This point needs more clarification.
- Line 244-245: While the model results suggested little to no variation in $\Delta^{17}O(HNO_3)$, many field observations indicated a clear diurnal trend (Vicars et al., 2013; Zhang et al., 2022). The author hypothesized that this was due to the relatively long chemical lifetime of $HNO_3$ in the system. However, as previously described in the Methods section, a dilution lifetime of 24 h was applied by the model, so it is unlikely that excessive $HNO_3$ would accumulate in the system, as shown in Figure 1. The 24 h dilution lifetime is actually lower than the typical $HNO_3$ lifetime in a realistic atmosphere (3-5 days). Please provide more detailed explanations.
- Line 250: I note that in Morin et al. (2011), the simulated $\Delta^{17}O(HO_2)$ was nonzero but close to 1-2 per mil, while in this work, the modeled $\Delta^{17}O(HO_2)$ was almost identical to 0 (visually from Figure 2 and Figure 4). The nonzero $\Delta^{17}O(HO_2)$ in Morin et al. (2011) was attributed to the $OH+O_3$ reaction. Does this $HO_2$ formation pathway significant in this work? This would impact the $\Delta^{17}O$ of $H_2O_2$, which is potentially useful for sulfate chemistry.
- Some of the reference citations are not consistent with the Copernicus publication style.

**References**

Blum, D. E., Walters, W. W., Eris, G., Takeuchi, M., Huey, L. G., Tanner, D., Xu, W., Rivera-Rios, J. C., Liu, F., Ng, N. L., and Hastings, M. G.: Collection of Nitrogen Dioxide for Nitrogen and Oxygen Isotope Determination—Laboratory and Environmental Chamber Evaluation, Analytical Chemistry, 95, 3371-3378, 10.1021/acs.analchem.2c04672, 2023.

Janssen, C., Guenther, J., Krankowsky, D., and Mauersberger, K.: Temperature dependence of ozone rate coefficients and isotopologue fractionation in 16O–1°oxygen mixtures, Chemical Physics Letters, 367, 34-38, https://doi.org/10.1016/S0009-2614(02)01665-2, 2003.

Krankowsky, D., Lämmerzahl, P., Mauersberger, K., Janssen, C., Tuzson, B., and Röckmann, T.: Stratospheric ozone isotope fractionations derived from collected samples, 112, https://doi.org/10.1029/2006JD007855, 2007.

Röckmann, T., Brenninkmeijer, C. A. M., Saueressig, G., Bergamaschi, P., Crowley, J. N., Fischer, H., and Crutzen, P. J.: Mass-Independent Oxygen Isotope Fractionation in Atmospheric CO as a Result of the Reaction CO + OH, 281, 544-546, doi:10.1126/science.281.5376.544, 1998.

Savarino, J., Vicars, W. C., Legrand, M., Preunkert, S., Jourdain, B., Frey, M. M., Kukui, A., Caillon, N., and Gil Roca, J.: Oxygen isotope mass balance of atmospheric nitrate at Dome C, East Antarctica, during the OPALE campaign, Atmos. Chem. Phys., 16, 2659-2673, 10.5194/acp-16-2659-2016, 2016.

Thiemens, M. H. and Jackson, T.: Pressure dependency for heavy isotope enhancement in ozone formation, 17, 717-719, https://doi.org/10.1029/GL017i006p00717, 1990.

Velivetskaya, T. A., Ignatiev, A. V., Budnitskiy, S. Y., Yakovenko, V. V., and Vysotskiy, S. V.: Mass-independent fractionation of oxygen isotopes during $H_2O_2$ formation by gas-phase discharge from water vapor, Geochimica et Cosmochimica Acta, 193, 54-65, https://doi.org/10.1016/j.gca.2016.08.008, 2016.

Vicars, W. C. and Savarino, J.: Quantitative constraints on the 17O-excess ($\Delta$17O) signature of surface ozone: Ambient measurements from 50°N to 50°S using the nitrite-coated filter technique, Geochimica et Cosmochimica Acta, 135, 270-287, https://doi.org/10.1016/j.gca.2014.03.023, 2014.

Vicars, W. C., Morin, S., Savarino, J., Wagner, N. L., Erbland, J., Vince, E., Martins, J. M. F., Lerner, B. M., Quinn, P. K., Coffman, D. J., Williams, E. J., and Brown, S. S.: Spatial and diurnal variability in reactive nitrogen oxide chemistry as reflected in the isotopic composition of atmospheric nitrate: Results from the CalNex 2010 field study, 118, 10,567-510,588, https://doi.org/10.1002/jgrd.50680, 2013.

Xu, H., Tsunogai, U., Nakagawa, F., Sato, K., and Tanimoto, H.: Diagnosing the stratospheric proportion in tropospheric ozone using triple oxygen isotopes as tracers, Atmos. Chem. Phys. Discuss., 2022, 1-20, 10.5194/acp-2021-1099, 2022.

Ye, C., Zhou, X., Pu, D., Stutz, J., Festa, J., Spolaor, M., Tsai, C., Cantrell, C., Mauldin, R. L., Campos, T., Weinheimer, A., Hornbrook, R. S., Apel, E. C., Guenther, A., Kaser, L., Yuan, B., Karl, T., Haggerty, J., Hall, S., Ullmann, K., Smith, J. N., Ortega, J., and Knote, C.: Rapid cycling of reactive nitrogen in the marine boundary layer, Nature, 532, 489-491, 10.1038/nature17195, 2016.

Zhang, Y.-L., Zhang, W., Fan, M.-Y., Li, J., Fang, H., Cao, F., Lin, Y.-C., Wilkins, B. P., Liu, X., Bao, M., Hong, Y., and Michalski, G.: A diurnal story of $\Delta$17O($\rm{NO}_{3}^{-}$) in urban Nanjing and its implication for nitrate aerosol formation, npj Climate and Atmospheric Science, 5, 50, 10.1038/s41612-022-00273-3, 2022.

---

## Author Comment (AC1)

**Response to Reviewers**

**Overall:** We appreciate the Reviewers' helpful and constructive feedback, which has helped to improve the manuscript significantly. Specifically, we have improved our discussion on $\Delta^{17}O(O_3^{term})$ values. We have provided additional reasoning behind using a constant $\Delta^{17}O(O_3^{term})$ value of 39.3±2‰ for simulating $\Delta^{17}O$ of $NO_y$ molecules for applications to the lower atmosphere using our newly developed gas-phase mechanism. We have further updated our paper to describe how oxygen mass-independent fractionation reactions other than $O_3$ formation could be adapted into our newly developed mechanism. However, we note that $O_3$ formation is the dominant source of mass-independent fractionation for the lower atmosphere, which is the targeted region of our mechanism. We have provided an additional comparison of various model predictions to field measurements, as suggested by Reviewer #1. We note that our model examples were not constructed to compare our model results to field observations quantitatively; thus, we have qualitatively highlighted the general trends captured or not captured with our new gas-phase chemical mechanism development compared to field $\Delta^{17}O$ observations of $NO_y$ molecules. This manuscript aims to demonstrate the development of a new gas-phase mechanism for tracking $\Delta^{17}O$ of $NO_y$ in the troposphere rather than to model and interpret previous $\Delta^{17}O(NO_y)$ field data, and this aim was supported by Reviewer #2. Finally, we have provided and documented our new gas-phase chemical mechanism in a supplemental file in addition to the previously available files on Zenodo and GitHub. Overall, these changes have improved the presented manuscript and increased the transparency and potential impact of the newly developed mechanism. A point-by-point response to all the reviewer comments is provided below.

**Chief Editor**

**Comment**: "Dear authors,
A short comment to highlight that the "Code and Data Availability" statement in your submitted manuscript must be changed in potential future versions. Currently, it says that the code is available in GitHub, which is incorrect and moreover, would not comply with our code and data policy. Actually, the repository, linked in the references is in Zenodo. Therefore, please, fix this in future versions, and include the link and DOI to the Zenodo repository in the "Code and Data Availability" section, not as a reference.
Regards
Juan A. Añel"

**Response**: Thank you, Dr. Añel, for catching this error. We initially uploaded our manuscript and posted the data and code on GitHub. However, we learned that this did not comply with the requirements of *GCD,* and we then posted our data/code in Zendo before our manuscript was posted for discussion. We now see that we forgot to change the "Code and Data Availability" statement before our work reached the discussion phase. In the revised manuscript, we have updated the "Code and Data Availability" statement to read: "The developed mechanism, box-model source codes, and the input and output files have been made publicly available at https://zenodo.org/records/10961373 (doi: 10.5281/zenodo.10961373)." This change was made on Lines 369-370.

**Reviewer #1**

**Comment**: In this paper, Walters et al. presented an isotopic chemical mechanism (ICOIN-RACM2) by incorporating oxygen isotopes into the gas-phase chemical mechanism RACM2. This new isotopic mechanism aims to simulate the oxygen mass- independent fractionation signals ($\Delta^{17}O$) of atmospheric oxidized reactive nitrogen ($NO_y$) and odd oxygen species. The authors adopted an isotope-tagging method by explicitly tracking the oxygen atom that was transferred from the terminal of ozone in each molecule. After examining the consistency between the new isotopic mechanism and the original RACM2 mechanism, the authors conducted two case studies by applying 0-D box model simulations. The first is to simulate the $\Delta^{17}O$ of $NO_y$ in ideal photochemical chamber experiments with different initial conditions, and the second is to simulate the diel cycles of $\Delta^{17}O$ in a summertime atmosphere. The author concluded that this new mechanism would help advance our understanding of $NO_y$ oxidation chemistry and is expected to be useful for future applications in 3-D CTMs. Oxygen isotope signals, especially $\Delta^{17}O$, have been widely used to constrain the oxidation chemistry of atmospheric reactive nitrogen. The novel isotopic tagging chemical mechanism introduced by this study could serve as a useful tool in such studies. I feel the paper is well written and the topic is of great interest for the readership of GMD. However, I have three major concerns on this study:

**Response:** We thank the reviewer for their feedback and consideration of our work. We have addressed all their raised concerns in a point-by-point response below.

**Major Comments:**

**Comment #1**: The assumption of a constant $\Delta^{17}O$ of ozone. The author cited the study of Vicars and Savarino et al. (2014) to validate the choice of a constant $\Delta^{17}O(O_3)$. I agree that the average tropospheric $\Delta^{17}O(O_3)$ from multiple observational studies is close to 26 ‰, but there are also significant seasonal cycles in these observations, including the Vicars and Savarino. (2014) study. For example, Savarino et al. (2016) suggested an ~10 ‰ variation in $\Delta^{17}O(O_3)$ in different seasons at Dome C. Xu et al. (2022) also indicated a >5 ‰ variation in the $\Delta^{17}O$ of the ozone terminal oxygen in Japan (Xu et al., 2022), which was attributed to a stratospheric intrusion event. These values are already large enough to impact the oxygen MIF transferred to other molecules from ozone. Moreover, it is well known that $\Delta^{17}O(O_3)$ is sensitive to both temperature and pressure (Thiemens and Jackson, 1990; Janssen et al., 2003) and that stratospheric ozone has a much greater $\Delta^{17}O$ than tropospheric ozone (Krankowsky et al., 2007). Therefore, adapting an average value from mostly near-surface observations is a major drawback of the method used in this study and severely limits the extension of the proposed mechanism to 3D CTMs, as the author suggested.

**Response**: We thank the reviewer for raising this important point, and we can see how the original discussion of our choice for $\Delta^{17}O(O_3^{term})$ could be perceived as oversimplified, particularly for users interested in applying the mechanism to environments other than near-surface conditions or close to normal, temperature, and pressure (NTP) conditions. Our model mechanism tracks the transfer of terminal O atoms of $O_3$ into $NO_y$ and $O_x$ species and does not calculate $\Delta^{17}O$ for these molecules directly. Therefore, depending on the environment (i.e., pressure and temperature), users may choose the appropriate $\Delta^{17}O(O_3^{term})$ value that best suits their environment/study. This provides users with a highly flexible model to probe how chemistry and/or $\Delta^{17}O(O_3^{term})$ may impact the $\Delta^{17}O$ in their applications. In our case, we demonstrate the newly developed mechanism for chamber simulations and under lower troposphere applications. Therefore, we have chosen a $\Delta^{17}O(O_3^{term})$ value of 39.3±2‰, consistent with near-surface observations (Vicars & Savarino, 2014). Further, this value was used for global nitrate

$\Delta^{17}O$ modeling using GEOS-Chem, which provided a reasonable match between observations and model (Alexander et al. 2020), so it should be reasonable for our model simulations to showcase the newly developed mechanism, as we discuss below.

In the revised manuscript, we have extended the introduction of the $\Delta^{17}O(O_3)$, $\Delta^{17}O(O_3^{term})$, and the reasoning behind using an assumed $\Delta^{17}O(O_3^{term})$ value of 39±2‰ for simulating $\Delta^{17}O$ transfer dynamics associated with $NO_y$ and $O_x$ molecules in the lower troposphere. The following lines were added to the revised manuscript on Lines 34-57 to highlight the variability in $\Delta^{17}O(O_3^{term})$, "In this work, we focus on the propagation of the oxygen isotope anomaly from $O_3$ mass-independent fractionation into $NO_y$ and $O_x$ molecules for applications to the lower atmosphere. The $\Delta^{17}O(O_3)$ has been measured to be between 20 and 46 ‰ (Krankowsky et al., 2000; Mauersberger et al., 2001). This range of values has been shown to track with the pressure and temperature associated with $O_3$ formation (Thiemens and Jackson, 1990; Morton et al., 1990). For typical tropospheric conditions, $O_3$ exhibits a $\Delta^{17}O$ between 20 and 30 ‰ (Johnston and Thiemens, 1997), with recent near-surface observations suggesting a mean $\Delta^{17}O(O_3)$ near 26 ‰ (Vicars and Savarino, 2014; Vicars et al., 2012; Ishino et al., 2017). $O_3$ is also isotopically asymmetrical such that the $\Delta^{17}O$ of its terminal and central O atoms are different (Janssen, 2005; Marcus, 2008). This intramolecular $\Delta^{17}O$ distribution is significant because the terminal O-atom of $O_3$ (defined as $O_3^{term}$) is preferentially transferred during oxidation reactions involving $O_3$ (Bhattacharya et al., 2008; Liu et al., 2001; Michalski and Bhattacharya, 2009; Walters and Michalski, 2016). The relationship between $\Delta^{17}O(O_3)$ and $\Delta^{17}O(O_3^{term})$ is complex, though experimental data has suggested the following relationship:
$$\Delta^{17}O(O_3^{term}) = 1.5 \times \Delta^{17}O(O_3) \qquad (1)$$

Applying this relationship to the assumed tropospheric mean $\Delta^{17}O(O_3)$ of 26‰ would imply a $\Delta^{17}O(O_3^{term})$ of 39 ‰, which is near the average of recent near-surface $\Delta^{17}O(O_3^{term})$ observations of 39.3±2‰ (Vicars and Savarino, 2014). It is important to note that there could be seasonal differences in $\Delta^{17}O(O_3^{term})$ as inferred from $\Delta^{17}O$ measurements of nitrate at Dome C (Savarino et al., 2016). On the other hand, direct observations of $\Delta^{17}O(O_3^{term})$ have reported insignificant seasonal variability at Dumont d'Urville (Ishino et al., 2017). Stratospheric intrusion events could introduce $O_3$ with an elevated $\Delta^{17}O(O_3^{term})$ due to higher stratosphere values relative to the troposphere (Krankowsky et al., 2007). Nevertheless, a recent modeling study of $\Delta^{17}O$ of atmospheric nitrate indicated that an assumed $\Delta^{17}O(O_3^{term})$ value of 39 ‰, reasonably reproduced global tropospheric observations (Alexander et al., 2020). Further, recent chamber simulations have reported a $\Delta^{17}O(NO_2)$ that reached as high as 40.1 ‰ (Blum et al., 2023), which is within the measurement uncertainty of the assumed $\Delta^{17}O(O_3^{term})$ value of 39.3 ±2‰, assuming $NO_2$ formation to be dominated by NO reaction with $O_3$. Thus, while there may be some unresolved uncertainty regarding the $\Delta^{17}O(O_3^{term})$ value, an assumed tropospheric average of 39.3±2 ‰, should reasonably approximate $\Delta^{17}O$ propagation into $NO_y$ molecules in the lower troposphere."

Further, in the method section, we highlighted how users can choose the $\Delta^{17}O(O_3^{term})$ value for their modeling scenarios and provided further elaboration for our choice of $\Delta^{17}O(O_3^{tem})$ value in our demonstration of the application of the mechanism. The following was added to Lines **123-131**, "For the demonstration of the developed mechanism for applications to chamber simulations and tropospheric chemistry, we have utilized a constant $\Delta^{17}O(O_3^{term})$ value of 39.3±2‰, based on near surface-level collections of $O_3$ on a nitrite coated filter (Vicars & Savarino, 2014; Ishino et al., 2017). This $\Delta^{17}O(O_3^{term})$ value was recently utilized in the global modeling of $\Delta^{17}O$ of atmospheric nitrate, demonstrating reasonable agreement between model simulation and observations of tropospheric nitrate (Alexander et al., 2020). The $\Delta^{17}O(O_3^{term})$ could have temporal variability as well as be

influenced by stratospheric intrusion events, which could introduce $O_3$ with a higher $\Delta^{17}O(O_3^{term})$ value. The developed model framework is highly flexible, and the user may apply a different $\Delta^{17}O(O_3^{term})$ than chosen for our model simulations, which will allow users to investigate both the chemical and $\Delta^{17}O(O_3^{term})$ variabilities on $\Delta^{17}O$ of $NO_y$ and $O_x$ species when interpreting field observations."

**Comment #2**: (2) The ignorance of other processes that could produce OMIF. There are several known chemical processes that can produce extra OMIF, e.g., the CO+OH reaction and the $HO_2+HO_2$ reaction (Röckmann et al., 1998; Velivetskaya et al., 2016), and perhaps additional unknown reactions exist. From my point of view, the isotopic tagging method can hardly handle these effects. The impact of these processes on the $\Delta^{17}O$ of atmospheric $NO_y$ could be small but should not be ignored in this study.

**Response**: We thank the reviewer for bringing up this excellent point. We note that our focus is on the lower atmosphere when applying the developed mechanism, in which ozone dominates the source of mass-independent fractionation. Therefore, we added the following to the revised manuscript on Lines 31-34, "While several atmospheric reactions can induce oxygen mass-independent fractionation (Röckmann et al., 1998; Velivetskaya et al., 2016), $O_3$ is the overwhelming source of mass-independent fractionation in the lower atmosphere, which derive from unconventional isotope effects during its formation (Gao and Marcus, 2001)."

We also note that we could easily track the OMIF associated with these reactions by creatively scaling the products associated with these reactions using our "Q" tagging approach. This highlights the incredible flexibility of our model framework without comprising computational resources by having to explicitly model every single O isotopologue of $NO_y$ and $O_x$ components, which would become particulary tedious when applied to several hundred reactions involving organics. This would be conducted by adjusting products to have a fraction of $Q$ (Eq. 3 of main text) that, once scaled by the assumed $\Delta^{17}O(O_3^{term})$ value of 39.3±2‰ (Eq. 2 of main text), would match the experimental results. For example, if we consider the $HO_2 + HO_2 \rightarrow H_2O_2 + O_2$ reaction may lead to a $\Delta^{17}O(H_2O_2)$ with an upper limit of 2.5‰ (Velivetskaya et al., 2016), we could adjust this reaction as follows:

$$HO_2 + HO_2 \rightarrow 0.94H_2O_2 + 0.032H_2OQ + 0.032H_2Q_2 + O_2$$

The fraction of "Q" would be = 0.0639, leading to a calculated $\Delta^{17}O$ of 2.5‰. In the revised manuscript, we added the following to Lines 136-145, "While our mechanism and application is focused on evaluating the propagation of oxygen isotope mass-independent fractionation from $O_3$ into $NO_y$ and $O_x$, the model could be adapted for tracking other potential oxygen mass-independent fractionation, such as $HO_2 + HO_2$ or $CO + OH$ reactions (Röckmann et al., 1998; Velivetskaya et al., 2016), by adjusting the product distribution of "Q" and "O", such that the fraction of "Q" once scaled by the chosen $\Delta^{17}O(O_3^{term})$ value would match the intended $\Delta^{17}O$ value associated with the oxygen mass-independent fractionation. Previous experiments have reported an increase in $\Delta^{17}O(H_2O_2)$ as the initial $O_2$ concentrations increased (Velivetskaya et al., 2016). This result was concluded to reflect the increased role of $O_3$ reactions in $H_2O_2$ formation, which is already tracked in our mechanism. The $CO + OH$ reaction, producing a $\Delta^{17}O$ in the residual CO would be extremely unlikely to affect the $\Delta^{17}O$ of $NO_y$ or $O_x$ due to the long atmospheric lifetime of CO relative to $NO_y$ or $O_x$. Therefore, we did not explicitly test these reactions' influence on $\Delta^{17}O$ of $NO_y$ or $O_x$ in this work but could easily be adapted in future iterations of the model."

**Comment #3:** Comparisons with observations to validate the proposed mechanism are lacking. Although the author conducted two different case simulations in this study, it is unclear why these cases were chosen. For the chamber simulations, it looks that the model configurations mimic the chamber experiments in Blum et al. (2023), but why the author didn't compare the model results with Blem et al. (2023)? This would provide the reader with a direct impression about the model performance. I noticed that in Blem et al. (2023), both $\Delta^{17}O(NO_2)$ and $\Delta^{17}O(NO_3^-)$ were measured, and the comparison should be straightforward. In addition, the measured $\Delta^{17}O(NO_2)$ in Blem et al. (2023) could exceed 40 ‰, which cannot be reproduced by assuming a constant 39.3 ‰ $\Delta^{17}O$ of terminal oxygen in ozone in this study. The modeled atmospheric $\Delta^{17}O(NO_2)$ trend in the latter case was also not consistent with recent observations (see my minor comment below). These simple comparisons suggested the limitations or incompetence of the current mechanism. Nevertheless, I appreciate much about the effort made by this study, and I look forward to incorporating this new oxygen isotopic mechanism into 3D CTMs in the future. I suggest the author to include direct comparisons with the existing observational data especially the chamber experiments, which would require a redesign of the box model simulations, and the author should make more explanations on the points I mentioned above in the revised manuscript. For these reasons I think a major revision is necessary before the final publication of this manuscript on GMD.

**Response:** We appreciate the suggestion for conducting a detailed comparison with the data reported in Blum et al., 2023; however, as pointed out by **Reviewer #2,** that was not the focus of this paper. We intend to simulate the data from the chamber experiments conducted by Blum et al., 2023, in a forthcoming paper. Combining both the new model mechanism development and simulating the chamber data led to an unfocused paper, warranting a separate paper. This was due to the level of detail that would be required to detail the model set-up to accurately simulate a chamber experiment (e.g., wall loss, experiment timing, etc.), while simultaneously describing a new chemical mechanism. Further, simulating the chamber data would provide additional insights into $\alpha$-pinene/$NO_x$ chemistry that would detract from the new chemical mechanism development. This type of paper and expected outcomes will be aimed at an audience focused on atmospheric chemistry, which, in our opinion, would not be appropriate for GMD, which has a broader research interest.

Here, we wanted to separately highlight the development of a new model mechanism for the explicit simulation of $\Delta^{17}O$ of $NO_x$-related chemistry. We also wanted to seek community feedback on this new mechanism and its development before applying it to chamber data. Therefore, the revised manuscript did not provide a detailed comparison with Blum et al., 2023.

The Reviewer makes an excellent point about the higher $\Delta^{17}O(NO_2)$ measured by Blum et al., 2023, of 40.1‰ compared to the considered $\Delta^{17}O(O_3^{term})$ value of 39.3‰. We note that in the revised manuscript, we considered the uncertainty ($\pm 1\sigma$) of $\Delta^{17}O(O_3^{term})$ of $\pm 2$‰ reported by Vicars et al., 2013, and we have updated our analysis and plots accordingly. Indeed, the measured $\Delta^{17}O(NO_2)$ of 40.1 ‰ is within the range $\pm 1\sigma$ for the $\Delta^{17}O(O_3^{term})$ value we have used in this work. In our updated discussion about the uncertainty in $\Delta^{17}O(O_3^{term})$ and our choice of using a set value of 39.3$\pm 2$‰ in response to **Comment #1**, we added the following to Lines 53-55, "Further, recent chamber simulations have reported a $\Delta^{17}O(NO_2)$ that reached as high as 40.1 ‰ (Blum et al., 2023), which is within the measurement uncertainty of the assumed $\Delta^{17}O(O_3^{term})$ value of 39.3 $\pm 2$‰, assuming $NO_2$ formation to be dominated by reaction with $O_3$."

**Minor Comments:**

**Comment #4**: I don't understand why the author used to added a parenthesis in the delta notation throughout the text. I suggest the author to follow the most conventional notation by using $\delta^{18}O$ and $\Delta^{17}O$.

**Response**: Thank you for raising this point. In previous submissions to EGU journals, we were instructed to add parentheses in the delta notation to comply with IUPAC recommendations. However, since the community appears to prefer the conventional notation of $\delta^{18}O$ and $\Delta^{17}O$ as opposed to $\delta(^{18}O)$ and $\Delta(^{17}O)$, we have updated our notation throughout the revised text.

**Comment #5**: Lines 47-48: Walters et al., 2018, Blum et al., 2020 and Chai and Hastings, 2018 didn't measure $\Delta^{17}O$ in their study and are improper citations here.

**Response**: Thank you for raising this point. We have removed references to these citations in the revised manuscript. For the referenced lines, we intended to highlight that recent methodological developments have expanded the community's ability to measure the oxygen isotope composition of various $NO_y$ molecules. Indeed, since the manuscript focuses on $\Delta^{17}O$, we can see how the referenced citations could be misleading.

**Comment #6**: Line 59-62: I suggest to rephrase this sentence. Clearly, photochemical equilibrium is no longer held at night because there is no light or photolysis of NO2, instead of nocturnal NO emission. I understand the author may want to describe that the freshly emitted NO would dilute the residual $\Delta^{17}O(NO_2)$ from the daytime, but the statement seems weird to me.

**Response**: Thank you for pointing out that this sentence was difficult to understand. We have revised this sentence as follows, "However, recent diel observations of $\delta^{18}O(NO_2)$ (which tracks with $\Delta^{17}O$) and $\Delta^{17}O(NO_2)$ reveal that this assumption is not universally valid due to substantial nocturnal NO emissions close to the surface (Walters et al., 2018; Albertin et al., 2021). The freshly emitted NO, with a presumed $\Delta^{17}O$ of 0 ‰, would dilute the residual $\Delta^{17}O$ of $NO_x$ from the daytime". These changes were made on Lines 80-83 in the revised manuscript.

**Comment #7**: Table 1: The $\Delta^{17}O(RO_2NO_2)$ should be $1/2*\Delta^{17}O(RO_2)$ plus $1/2*\Delta^{17}O(RO_2)$.

**Response**: Thank you for the comment; however, we have calculated $\Delta^{17}O(RO_2NO_2)$ from the nitro (-$NO_3$) group. We think this is reasonable because there are likely ways to collect $RO_2NO_2$ and hydrolyze the $NO_3$ group. Thus, the mass-balance equation in the original manuscript is correct. We attempted to include a table endnote to point out our assumption, but we incorrectly labeled the table footnote (as "a"), while we had a superscript next to $RO_2NO_2$ as "*". We have corrected our table footnote in the revised manuscript (Table 1).

**Comment #8**: Section 2.1: Does the model incorporate the photolysis of particle nitrate? The author mentioned the importance of this process but it's not clear if it is included by the model. In addition, I suggest the author to add a table that describing the reaction mechanism in a supplemental file, otherwise the reader not familiar with the RACM2 mechanism would have to search the model code, which is a rather tough task.

**Response**: Thank you for raising this point. The model does not include photolysis of particulate nitrate because our model is a gas-phase mechanism and does not model particulate nitrate. As we mentioned in the original manuscript, this could be a potential limitation for simulating $\Delta^{17}O(HONO)$. In the revised manuscript, we added the following to Lines **153-158**, "Since the ICOIN-RACM2 mechanism does not model particulate nitrate, we cannot model its photolysis, which could limit our ability to simulate $\Delta^{17}O(HONO)$. Additionally, our gas-phase mechanism does not include $NO_2$

heterogeneous reactions, which could also be an important source of HONO (Chai et al., 2021). Users interested in accurately simulating $\Delta^{17}O(HONO)$ may need to consider adding relevant reactions. Still, a future comparison between $\Delta^{17}O(HONO)$ observations and model simulations based on the ICOIN-RACM2 framework should provide pivotal insight into HONO formation."

Further, we have added a supplemental file of the ICOIN-RACM2 mechanism in the revised submission. This was also requested by **Reviewer #2**.

**Comment #9:** Figure 1 and Figure 2: I suggest to change the y scales in different subplots.
**Response:** Thank you for the suggestion. We have updated these figures in the revised manuscript so that the subplots have different y-scales. This change improves the visualization of the model simulations.

**Comment #10:** Line 201-202: The simulated $\Delta^{17}O$ of NO was described in this sentence but not shown in the figure. Please add it.
**Response:** Thank you for raising this point. In the revised manuscript, we have added the $\Delta^{17}O(NO)$ simulations to Fig. 2 and moved the $\Delta 17O$ of ONIT and its formation mechanism to a separate plot (Fig. 3) to maintain consistency between Fig. 2 and Fig. 4.

**Comment #11:** Line 297-298: Another possible explanation for the nonzero $\Delta^{17}O$ of NO is the isotopic exchange between NO and $NO_2$. The author should easily test this hypothesis by turning off the NO-$NO_2$ exchange reaction in the mechanism.
**Response:** Thank you for this suggestion. We added to the discussion of the potential for NO/$NO_2$ isotope exchange in the revised manuscript on Lines 277-284, "During the daytime, the simulated $\Delta^{17}O(NO_2) \approx \Delta^{17}O(NO)$, due to due to the rapid $NO_x$ photochemical cycling. However, during the nighttime, $\Delta^{17}O(NO_2)$ was greater than $\Delta^{17}O(NO)$ due to the role of nighttime NO emissions with an assumed $\Delta 17O(NO) = 0$ ‰. While NO and $NO_2$ isotope exchange would lead to $\Delta^{17}O(NO) = \Delta^{17}O(NO_2)$, its role in influencing $\Delta^{17}O$ depends on the concentrations of NO and $NO_2$, as previously discussed for $\delta^{15}N$ of $NO_x$ (Walters et al., 2016). In the diel model simulations, nighttime NO concentrations were less than 0.1 ppb (Fig. 1) due to its titration by $O_3$. Under these conditions, the rate of NO/$NO_2$ isotope exchange was slow relative to NO oxidation or the rate of NO primary emission, leading to a low nighttime $\Delta^{17}O(NO)$ value for the simulation conditions of low nighttime $NO_x$ relative to $O_3$ concentrations."

**Comment #12:** Line 298-234: While the overall day high and night low trend in $\Delta^{17}O(NO_2)$ was reproduced by the model, the simulated diurnal $\Delta^{17}O(NO_2)$ trend was indeed not consistent with the observations in Albertin et al. (2021). Albertin et al. (2021) reported that the maximum $\Delta^{17}O(NO_2)$ occurred at noon, while the model suggested a minimum at noon. New observations by Albertin et al. (2024) indicated a similar trend. This point deserves further discussion. Moreover, in Figure 4, the simulated diurnal $\Delta^{17}O(NO_2)$ seems to be slightly greater than $\Delta^{17}O(NO)$, especially in case 20. This is astonishing since both photochemical cycling and isotopic exchanges tend to force them equal. Please explain it.
**Response:** Thank you for pointing this out. First, we want to point out that diel simulations, shown to demonstrate the ICOIN-RACM2 mechanism, represent two simple case scenarios, and we do not claim them to be representative of all environments that will have different meteorological conditions and emissions than the test cases. Thus, we added the following to the revised manuscript on Lines 213-218, "The diel simulations are used to demonstrate the utility of the ICOIN-RACM2 mechanism. The two near-surface summertime model scenarios do not represent all

atmospheric conditions, including meteorology, actinic flux, and emission rates, which will influence the model $\Delta^{17}O$ values. Thus, these simulations cannot quantitatively be compared with various field $\Delta^{17}O$ data. This type of comparison would require a more targeted simulation set-up to represent the atmospheric conditions at a particular site. Still, we have compared qualitative trends predicted with the diel simulations with some available field data of $\Delta^{17}O$ observations."

Nonetheless, perhaps our initial discussion of the $\Delta^{17}O(NO_2)$ diel simulations relative to available data was oversimplified. We note that in Albertin et al., 2021, the highest $\Delta^{17}O(NO_2)$ value was from sample collection from 9:00 am to 12:00 pm, which would include collection of $NO_2$ during a period in which the model simulation for $\Delta^{17}O(NO_2)$ is near its peak, such that our simulations are not different from the observations. We also note that the simulated $\Delta^{17}O(NO_2)$ values are sensitive to meteorological conditions, and our simulations are for summertime conditions, while observations in Albertin et al., 2021 were for springtime conditions. We are not necessarily trying to simulate the data in Albertin et al., 2021; we only wish to point out that our model is consistent with trends of available oxygen isotope data of $NO_2$. In the revised manuscript, we changed this discussion as follows on Lines 285-300, "The predicted $NO_2$ diurnal cycles of elevated $\Delta^{17}O$ during the daytime and low $\Delta^{17}O$ during the nights are generally consistent with summertime $\delta^{18}O$ observations (which track with $\Delta^{17}O$) in West Lafayette, IN, US (Walters et al., 2018), and recent diel observations of $\Delta^{17}O$ at Grenoble, FR, during the spring (Albertin et al., 2021). However, there are some slight differences in the daytime $\Delta^{17}O(NO_2)$ observations compared to the model simulations, in which the highest $\Delta^{17}O(NO_2)$ occurred for samples collected between 9 am – 12 pm (Albertin et al., 2021). In comparison, the model indicated the highest $\Delta^{17}O(NO_2)$ around 6 to 8 am following the return of photolysis near sunrise. The observations indicate a subsequent daytime decay of $\Delta^{17}O(NO_2)$ (Albertin et al., 2021). The model also indicates a daytime decay in $\Delta^{17}O(NO_2)$ following the initial maximum $\Delta^{17}O(NO_2)$ that coincides with the onset of photolysis; however, the model expects an increase in $\Delta^{17}O(NO_2)$ in the late afternoon due to increased $O_3/HO_x$ levels from the decrease in actinic flux. We do not intend to accurately simulate the previously reported $\Delta^{17}O(NO_2)$ values (Albertin et al., 2021). Some of the nuanced differences between the model simulation and observations of $\Delta^{17}O$ are likely due to differences in meteorological conditions, as the model was simulated for summertime while the observations were from springtime and for a different latitude and longitude. Further, our model neglects transport and assumes a constant emission rate, which could influence the diel $\Delta^{17}O(NO_2)$ predictions. Nevertheless, the ICOIN-RACM2 mechanism appears to capture the general diurnal trend of $\Delta^{17}O(NO_2)$. We envision that future adaptation of the ICOIN-RACM2 mechanism into a chemical transport model would provide useful insight for constraining $NO_x$ photochemical cycling based on a comparison to field $\Delta^{17}O(NO_2)$ measurements."

Additionally, we added to the discussion section the differences between $\Delta^{17}O(NO)$ and $\Delta^{17}O(NO_2)$, as described in our response to **Comment #11**, "During the daytime, the simulated $\Delta^{17}O(NO_2) \approx \Delta^{17}O(NO)$, due to due to the rapid $NO_x$ photochemical cycling. However, during the nighttime, $\Delta^{17}O(NO_2)$ was greater than $\Delta^{17}O(NO)$ due to the role of nighttime NO emissions with an assumed $\Delta^{17}O(NO) = 0$ ‰. While NO and $NO_2$ isotope exchange would lead to $\Delta^{17}O(NO) = \Delta^{17}O(NO_2)$, its role in influencing $\Delta^{17}O$ depends on the concentrations of NO and $NO_2$, as previously discussed for $\delta^{15}N$ of $NO_x$ (Walters et al., 2016). In the diel model simulations, nighttime NO concentrations were less than 0.1 ppb (Fig. 1) due to its titration by $O_3$. Under these conditions, the rate of $NO/NO_2$ isotope exchange was slow relative to NO oxidation or the rate of NO primary emission, leading to a low nighttime $\Delta^{17}O(NO)$ value for the simulation conditions of low nighttime $NO_x$ relative $O_3$ concentrations." These changes were made on Lines 277-284 in the revised manuscript.

**Comment #13:** Line 235: The discussion of the diurnal trend in $\Delta^{17}O(HONO)$ needs more attention because the photolysis of p-$NO_3^-$ seems to not be included by the model, which is however crucial to the HONO budget in many environments (e.g., (Ye et al., 2016)). This point needs more clarification.

**Response:** We thank the reviewer for pointing this out. As we noted in section 2.1 as well in **Comment #8**, our model is a gas-phase mechanism and does not include photolysis of $pNO_3$ or heterogenous reactions of $NO_2$, which could be an important source of HONO. We added the following to the revised manuscript on Lines 153-158, "Since the ICOIN-RACM2 mechanism does not model particulate nitrate, we cannot model its photolysis, which could limit our ability to simulate $\Delta^{17}O(HONO)$. Additionally, our gas-phase mechanism does not include $NO_2$ heterogeneous reactions, which could also be an important source of HONO (Chai et al., 2021). Users interested in accurately simulating $\Delta^{17}O(HONO)$ may need to consider adding relevant reactions. Still, a future comparison between $\Delta^{17}O(HONO)$ observations and model simulations based on the ICOIN-RACM2 framework should provide pivotal insight into HONO formation."

**Comment #14:** Line 244-245: While the model results suggested little to no variation in $\Delta^{17}O(HNO_3)$, many field observations indicated a clear diurnal trend (Vicars et al., 2013; Zhang et al., 2022). The author hypothesized that this was due to the relatively long chemical lifetime of $HNO_3$ in the system. However, as previously described in the Methods section, a dilution lifetime of 24 h was applied by the model, so it is unlikely that excessive $HNO_3$ would accumulate in the system, as shown in Figure 1. The 24 h dilution lifetime is actually lower than the typical $HNO_3$ lifetime in a realistic atmosphere (3-5 days). Please provide more detailed explanations.

**Response:** Thank you for raising this point. First, we would like to reiterate that our goal is not to explicitly compare with all types of field data from various atmospheric conditions, as noted in response to **Comment #12**. Additionally, our model simulates $\Delta^{17}O$ of $HNO_3$, while the field data of diurnal variability primarily refers to $pNO_3$. A recent observational study has shown that $\Delta^{17}O(HNO_3)$ does not necessarily equal $\Delta^{17}O(pNO_3)$ (Kim et al., 2023). Assuming that their diurnal variability might be expected to be similar, we have qualitatively compared the simulation prediction with the finding of some reports of diurnal $\Delta^{17}O(pNO_3)$ variability, "While the modeled diel $\Delta^{17}O(HNO_3)$ indicated no substantial diurnal variations, several field studies have indicated significant diurnal variability of $\Delta^{17}O(pNO_3)$ in polluted mega-cities (Zhang et al., 2022), as well as off the coast of California (Vicars et al., 2013). Commonly, $\Delta^{17}O(HNO_3)$ is thought to be equal to $\Delta^{17}O(pNO_3)$ due to the thermodynamic equilibrium between $HNO_3$ and $pNO_3$ (Alexander et al., 2009). However, recent data would suggest that $\Delta^{17}O(HNO_3)$ may not always be equal to $\Delta^{17}O(pNO_3)$ due to contributions of $pNO_3$ in the coarse aerosol phase that may not achieve thermodynamic equilibrium with $HNO_3$ (Kim et al.,2023). If we consider that the $\Delta^{17}O(pNO_3)$ diurnal variability should follow $\Delta^{17}O(HNO_3)$, the discrepancy between model and observations of diurnal variability would suggest that the lifetime of $pNO_3$ in these previous studies must be shorter than predicted in our model for $HNO_3$. Our model simulation was conducted using a gas-phase mechanism within a simple box-model framework. Potentially important $pNO_3$ loss processes not included in our model include $pNO_3$ photolysis and wet/dry deposition. These processes should not alter the $\Delta^{17}O$ of $pNO_3$ but could reduce the lifetime of $pNO_3$, leading to a significant diurnal variation in $\Delta^{17}O$. Additionally, our model simulation does not include transport or changes in boundary layer height and break up of the nocturnal boundary layer, which could also influence $\Delta^{17}O$ diurnal variations of $HNO_3$ and $pNO_3$." These changes were made on Lines 318-331 in the revised manuscript.

**Comment #15:** Line 250: I note that in Morin et al. (2011), the simulated $\Delta^{17}O(HO_2)$ was nonzero but close to 1-2 per mil, while in this work, the modeled $\Delta^{17}O(HO_2)$ was almost identical to 0 (visually from Figure 2 and Figure 4). The nonzero $\Delta^{17}O(HO_2)$ in Morin et al. (2011) was attributed to the $OH+O_3$ reaction. Does this $HO_2$ formation pathway significant in this work? This would impact the $\Delta^{17}O$ of $H_2O_2$, which is potentially useful for sulfate chemistry.

**Response:** Thank you for raising this point. This difference is due to including O isotope exchange reactions with $HO_2$ and $O_2$ from Lyons, 2001 in our model. In the revised manuscript, we conducted additional simulations in which we turned off the $HO_2/O_2$ isotope exchange, and the model simulated a non-zero $\Delta^{17}O(HO_2)$ value ranging from 0 up to 3 ‰ depending on the model conditions, consistent with Morin et al., 2011. We added the following discussion to the revised manuscript, "We note that the simulated $\Delta^{17}O(HO_2)$ was lower than previous $\Delta^{17}O(HO_2)$ simulations (Morin et al., 2011), which tended to be between 1 to 2 ‰. This difference is because we have included oxygen isotope exchange reactions involving $O_2$ and $HO_2$ (Lyons, 2001) (i.e., O-Exchange13 and O-Exchange14 in Table 2) in the ICOIN-RACM2 mechanism, which rapidly remove $\Delta^{17}O > 0$ ‰ in the generated $HO_2$. Without including this oxygen isotope exchange reaction, the ICOIN-RACM2 modeled $\Delta^{17}O(HO_2)$ predicts a non-zero $\Delta^{17}O(HO_2)$ that can be as high as 3 ‰ dependent on the model conditions (Fig. 5), consistent with previous model simulations (Morin et al., 2011). While the $\Delta^{17}O(HO_2)$ is expected to have a minor impact on the $\Delta^{17}O$ of $NO_y$ species (Alexander et al., 2009), we should consider the importance of the role of oxygen isotope exchange between $O_2$ and $HO_2$ influencing $\Delta^{17}O(HO_2)$, as it will be an important source of $\Delta^{17}O$ of $H_2O_2$, which is propagated into atmospheric sulfate (Savarino et al., 2000)." These changes were made on Lines 332-340 in the revised manuscript.

**Comment:** Some of the reference citations are not consistent with the Copernicus publication style
**Response:** Thank you for pointing this out. We have checked and updated our references in the revised manuscript as needed.

**Reviewer #2:**

**Comment:** Overall, I found the paper to be very well written and found that the novel isotopic tagging chemical mechanism developed in this study could serve as a very useful tool in the future to evaluate and understand NOy chemistry and is of great interest for the readership of GMD. However, I did have several major concerns that should be addressed before final publication:
**Response:** We thank the reviewer for their feedback on our manuscript and for recognizing the novel development of our mechanism, which we hope will be useful for future modeling efforts. We have addressed all the reviewers' comments, and below, you will find a point-by-point response to the raised concerns.

**Comment #1:** I could not find the mechanism files on GitHub and at minimum a link of where to find these files should be included in the text. They should be indexed with a DOI for permanence /reference and should be accessible for reviewers to see before publication. Currently, the unavailability of these files means the work is not reproduceable and is not in compliance with GMD's data policy.
**Response:** We thank the reviewer for raising this point. We intended to publicize our model simulations' mechanism, input, and output files before the review process. They were initially posted on GitHub and to Zenodo with a DOI; however, the original version of the manuscript did not clearly state our reference to Zendo with the associated DOI. We have updated our Date and Code Availability statement in the revised manuscript to clarify where the files associated with this

manuscript can be accessed.  This update was made on Lines 369-370 in the revised manuscript, "The developed mechanism, box-model source codes, and the input and output files have been made publicly available at https://zenodo.org/records/10961373 (doi: 10.5281/zenodo.10961373)."

**Comment #2:**  The mechanism developed adds 55 new species and 727 replicate reactions to RACM2. In order for these improvements to more easily be incorporated into other box model mechanisms available in F0AM (that may treat BVOC reactions with NOy differently than RACM2), or for these improvements to more easily make their way into CTMs as the authors suggest, I firmly believe the authors need to include as a supplement a table with a complete listing of the 55 species added including their SMILES or InChI codes like is done in the Bates et al., 2022 ACPD supplement (https://doi.org/10.5194/acp-22-1467-2022-supplement).  Including this information will allow anyone wanting to take these reactions w/ new species and map them to chemical mechanisms other than RACM2 to their mechanism in an automated way that may lump compounds in different ways (e.g. using python's RDKit library).  Pye et al., 2023 (https://acp.copernicus.org/articles/23/5043/2023/) makes an incredibly strong case that when SMILES or InChI codes available with new chemical mechanisms, end users who may want to take what's developed and incorporate it into other mechanisms is much easier to do in an automated way using RDKit. This is *especially* true for large mechanisms where doing this by hand is extraordinarily tedious. I specifically recommend the authors include the InChI codes rather than SMILES codes because they are better at taking into account isomerization and isotopes in a way that SMILES codes do not. Using RD-Kit, end users can transform the new species into molecules when InChI codes or SMILES codes are provided, thus enabling automated comparisons to other mechanisms (e.g. like MCM, which also provides these codes), CMAQ (Pye et al., 2023), and  GEOS-Chem, which is moving to also incorporate these codes currently in their mechanism metadata. Thus, for the more widespread adoption of this tool, and for it to truly have the impact the authors are hoping, I really think its extremely critical for the authors to provide these with the paper in a supplement. I know this may seem tedious especially for small inorganic molecules, but the value add for future users by providing this is quite critical for mechanisms of this size.

**Response:**  Thank you for this comment. We agree that providing a detailed mechanism and species table would help expand the impact of our newly developed mechanism.  Therefore, in the revised manuscript, we have provided a detailed **supplement file** that documents the ICOIN-RACM2 species, photolysis reactions, thermal reactions, troe reaction parameters, troe equilibrium parameters, and special rate expressions.  This file also contains the heterogeneous reaction additions used for the diel simulations.   We appreciate the reviewer's suggestion about providing SMILES and lnChl codes for the new species utilized in our mechanism.  However, the new species we have created to simulate $\Delta^{17}O$ transfer and propagation from $O_3$ are not real molecules.  Instead, these new species are tracers, so they do not have SMILES or lnChl codes based on our understanding of these classification systems. Further, the base RACM2 mechanism is a lumped mechanism, and some species in the mechanism refer to a category of molecules rather than a specific molecule.  This also prevents us from providing these codes. The new supplement file that details the mechanism is available in addition to the MATLAB files and output, which should make our model mechanism transparent.  Further, we are open to collaboration if other researchers are interested in using our model framework in other chemical mechanisms.

**Comment #3:** The other reviewer raises a really critical point that the average tropospheric d17O(O3) is 26‰, but the authors currently don't take into account known variations from this average.  To address this point, I think there should at least be some discussion of the known variations away from this in the main text. I agree this assumption certainly limits the extension of the mechanism for

3D-CTMS and suggest that if that it would be very valuable for future users of the mechanism if there was some built in way to account for this variation within the mechanism (perhaps by including other +/-% values with suggestions of when to use as a comment in the mechanism for the relevant rxns?). At minimum some discussion is needed about this in the main text.

**Response:** We thank the reviewer for raising this point. Indeed, we possibly oversimplified our discussion/rationale of $\Delta^{17}O(O_3^{term})$, which would be important in modeling $\Delta^{17}O$ of $NO_y$ and $O_x$ species. Since our focus was on applications to chamber experiments under normal, temperature, and pressure (NTP) conditions as well as simulating tropospheric chemistry, we had chosen a $\Delta^{17}O(O_3^{term})$ value of 39.3±2‰. Similarly to our response to **Reviewer #1, Comment #1**, we have expanded our discussion on the variability of $\Delta^{17}O(O_3^{term})$ and provided additional support for our reason for using a fixed $\Delta^{17}O(O_3^{term})$ value of 39.3±2‰ for demonstrating the potential utility of our mechanism. We have also noted that while our model does not simulate the $\Delta^{17}O(O_3^{term})$ values, users can select the $\Delta^{17}O(O_3^{term})$ value in their offline calculations of $\Delta^{17}O$ values of $NO_y$ and $O_x$ species. These additional discussions were added to the revised manuscript on Lines 31-57 and Lines 123-130.

**Comment #4:** For the Editors/Authors: I didn't find the lack of comparison to observational data as problematic as the other reviewer. I don't think the purpose of this paper was to actually interpret observations, but merely provide a tool for people to do so in the future. To that end, I think the choice of submitting to GMD was entirely appropriate and that it is not necessary for its publication to include that analysis here. I believe that is an entirely different paper for an entirely different journal, but that this work simply provides the tools necessary to enable others to do that sort of work.

**Response:** We thank the reviewer for providing this comment and supporting our decision not to compare our simulations with detailed chamber experiments reported in Blum et al., 2023. Due to the extensive work developing the mechanism and its potential to be applied in chemical transport models, we wanted to publish it separately from the chamber experiments. A paper comparing chamber experiments and model simulations will be presented soon.